



**The Role of Internal Variability in Regional Climate Change**
Clara Deser* and Adam S. Phillips
National Center for Atmospheric Research, Boulder CO USA
*EGU Nonlinear Processes in Geophysics Special Issue*
"Interdisciplinary perspectives on climate sciences – highlighting past and current scientific
achievements"
* Corresponding author: Clara Deser cdeser@ucar.edu





**Abstract**
Disentangling the effects of internal variability and anthropogenic forcing on regional climate
change remains a key challenge with far-reaching implications. Due to its largely unpredictable
nature on timescales longer than a decade, internal climate variability limits the accuracy of climate
model projections, introduces challenges in attributing past climate changes, and complicates
climate model evaluation. Here, we highlight recent advances in climate modeling and physical
understanding that have led to novel insights on these key issues. In particular, we synthesize new
findings from Large Ensemble simulations with Earth System Models, Observational Large
Ensembles, and "dynamical adjustment" methodologies, with a focus on European climate.
**1. Introduction**
*a. Internal variability and forced climate change*
The climate system is highly variable in both space and time. This variability originates from
processes within the coupled ocean-atmosphere-cryosphere-land-biosphere system, as well as
from external influences such as solar and orbital cycles, volcanic eruptions, and anthropogenic
emissions of greenhouse gases and sulfate aerosols. A primary source of internally-generated
variability is the atmospheric general circulation, which produces familiar day-to-day and week-
to-week weather fluctuations. The non-linear nature of atmospheric dynamics limits predictability
to less than a few weeks; beyond this time scale, atmospheric motions may be considered as
random stochastic processes, often termed "weather noise" (e.g., Lorenz, 1963; Leith, 1973; James
and James, 1992). It is important to note that such "weather noise" imparts variability on a
continuum of time scales, from sub-monthly to decadal and longer (e.g., Madden, 1975; Deser et
al. 2012; Thompson et al. 2015).



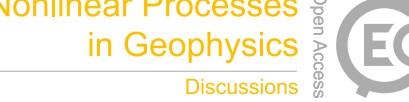

Another important source of internally-generated variability is the coupling between the ocean and
atmosphere. Large-scale air-sea interactions give rise to distinctive patterns (or "modes") of
variability on interannual and longer time scales, including phenomena such as "El Niño –
Southern Oscillation" (ENSO; Wang et al. 2017), "Pacific Decadal Variability" (PDV; Newman
et al. 2016) and "Atlantic Multi-decadal Variability" (AMV; Zhang et al. 2019). Like the
atmospheric general circulation, these coupled modes are governed by non-linear dynamical
processes which limit their predictability.  For example, forecast skill is generally limited to 1-2
years for ENSO (Jin et al., 2008; DiNezio et al. 2017; Wu et al. 2021), 5 years for PDV (Teng and
Branstator, 2010; Meehl et al., 2016; Gordon and Barnes, 2022) and 10 years for AMV (Griffies
and Bryan, 1997; Trenary and DelSole, 2016; Yeager et al., 2020). Beyond these predictability
time horizons, internally-generated variability can be thought of as a "roll of the dice", introducing
unavoidable uncertainty to climate model projections especially at local and regional scales (e.g.,
Deser et al. 2012, 2014 and 2020).

Not only does unpredictable internal variability cause irreducible uncertainty in future climate
projections, it also confounds interpretation of the historical climate record. For example, internal
variability may partially obscure the regional climate response to external forcings including
industrial greenhouse gas emissions, stratospheric ozone depletion and volcanic eruptions
(Wallace et al., 2013; Schneider et al. 2015; Lehner et al. 2016; McGraw et al. 2016).  In some
areas, climate trends driven by internal processes may even outweigh those due to anthropogenic
influences over the past 30-60 years (Deser et al., 2012, 2016 and 2017; Wallace et al., 2013; Swart
et al. 2015; Lehner et al. 2017). The co-existence of internal and anthropogenic influences





necessitates a probabilistic approach to detection and attribution of the human contribution to
anomalous weather and climate events.

The prevalence of internal climate variability also complicates model evaluation efforts, since the
simulated temporal sequence of (unpredictable) internal variability need not match observations
even if the model's physics are realistic. Further, the brevity of the instrumental record provides
only a limited sampling of internal variability, hindering robust model evaluation. Thus, climate
models may show an apparent bias with respect to observations, but this could be entirely
attributable to sampling issues rather than indicative of a true bias due to incorrect model physics.
Apparent model bias due to sampling uncertainty must be kept in mind when assessing fidelity of
simulated modes of internal variability (e.g., Wittenberg et al. 20xx; Deser et al. 2017; Capotondi
et al. 2020; Fasullo et al. 2021; McKenna and Maycock, 2021), transient climate sensitivity (Dong
et al. 2021; Andrews et al. 2022), and "signal-to-noise" properties of initial-value predictions and
forced responses (e.g., Scaife and Smith, 2018; Smith et al., 2020; Klavans et al. 2021). In
particular, even with 100 years of data, sampling uncertainty is a limiting factor for evaluating
ENSO properties in climate models, including its global atmospheric teleconnections and
associated climate impacts (Deser et al. 2017 and 2018; Capotondi et al. 2020) and forced changes
thereof (Stevenson et al. 2012; Maher et al. 2018; Maher et al. 2022; O'Brien and Deser, 2022).
This issue is particularly acute for model assessment of modes of decadal variability such as PDV
and AMV due to the paucity of samples in the short instrumental record (Deser and Phillips 2021;
Fasullo et al. 2021).

***b. Initial-condition Large Ensemble Simulations with Earth System Models***



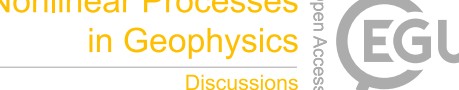

To overcome the issue of sampling uncertainty, a recent thrust in climate modeling is to run a large
number of simulations (30-100) with the same coupled model and the same radiative forcing
protocol (historical and/or future scenario) but vary the initial conditions.  The initial-condition
variation can be accomplished by introducing a random perturbation to the atmosphere on the order
of the model's numerical round-off error (e.g., $10^{-14}$ K in the case of atmospheric temperatures;
Kay et al. 2015) or it can be done by selecting a different ocean state from a long control run of
the coupled model, or a combination of the two (Deser et al. 2020 and Rodgers et al. 2021).
Regardless of the method used, the initial-condition perturbation serves to create ensemble spread
once the memory of the initial state is lost, typically within a month for the atmosphere and a few
years to a couple of decades for the ocean. The ensuing ensemble spread is thus solely attributable
to random internal variability (e.g., the "butterfly effect" in chaos theory); see Lorenz (1963) and
Tel et al. (2019). Because the temporal sequences of internal variability unfold differently in the
various ensemble members once the memory of the initial conditions is lost, one can estimate the
forced component at each time step (at each location) by averaging the members together, provided
the ensemble size is sufficiently large. The internal component in each ensemble member is then
obtained as a residual from the ensemble-mean. Note that a larger ensemble may be needed for
some aspects of the forced response than others: for example, forced changes in ocean heat content
may be readily detected with just a few members, while forced changes in the characteristics of
internal variability may require a much larger ensemble (Milinski et al., 2020).

Initial-condition Large Ensembles (LEs for short) have proven enormously useful for separating
internal variability and forced climate change on regional scales in models, and for providing
robust sampling of models' internal variability by pooling together all of the ensemble members



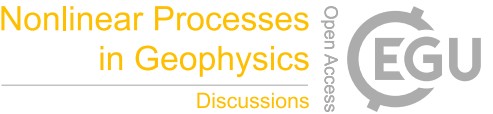

(e.g., Deser et al., 2012; Kay et al., 2015; Maher et al., 2019; Deser et al. 2020; Lehner et al., 2020).
They have also been used to assess externally-forced changes in the characteristics of simulated
internal variability, including extreme events for which large sample sizes are crucial (e.g., Tebaldi
et al., 2021; O'Brien and Deser, 2022). Additionally, they have served as methodological testbeds
for evaluating approaches to detection and attribution of anthropogenic climate change in the
(single) observational record (e.g., Deser et al., 2016; Barnes et al., 2019; Sippel et al., 2019 and
2021; Santer et al. 2019; Bonfils et al., 2019; Wills et al., 2020). Until the advent of LEs, it was
problematic to identify the sources of model differences in the Coupled Model Intercomparison
Project (CMIP) archives due to the limited number of simulations (generally < 3) for each model
(i.e., structural uncertainty was confounded with uncertainty due to internal variability). This
concern has been largely alleviated thanks to the recent availability of LEs with multiple earth
system models (e.g., Deser et al. 2020; Lehner et al., 2020).

***c. Observationally-based Large Ensemble***
Just as in a model LE, the sequence of internal variability in the real world could have unfolded
differently. That is, the observational record traces only one of many possible climate histories that
could have happened under the same external radiative forcing. For example, El Niño and La Niña
events could have occurred in a different set of years, and positive or negative regimes of PDV
and AMV could have taken place in different decades. This concept of alternate chronologies,
sometimes referred to as the "Theory of Parallel Climate Realizations" (Tel et al., 2019) or the
notion of "Contingency" (Gould, 1989), has major implications that call for a reframing of
perspective. For example, it means that a single model simulation of the historical period need not
match the observed record, even if the model is "perfect" in its physical representation of the real



world's climate. However, the statistical characteristics of the model's internal variability must
agree with those of the real world, taking into account sampling uncertainty (uncertainty due to
limited sampling in the short observational record). Thus, while a single ensemble member need
not match observations, the ensemble as a whole should encompass the instrumental data, provided
there are enough members to adequately span the range of possible sequences of internal variability
(Suarez-Guttierez et al. 2021).

Another implication of the concept of "parallel climate realizations" is that the climate trends we
have experienced are not unique. In analogy with a model LE, the observational record is just one
"member" of a larger set of possible "members". Although one cannot replay the "tape of history",
one can construct an "Observational LE" by generating alternate synthetic sequences of internal
variability from the instrumental data. Conceptually, this involves removing an estimate of the
forced component from the data and then randomizing the residual (internal) variability in time.
Importantly, the randomization procedure must be done in a way that preserves the statistical
properties of the observed variability including its variance, temporal autocorrelation, and spatial
patterns. The resulting synthetic sequences of internal variability derived from the observational
record can then be added back to the time-evolving forced response obtained from a climate model
LE.

The development of statistically-based Observational LEs is just beginning, with recent efforts
targeting surface climate fields (McKinnon et al., 2017; McKinnon and Deser, 2018 and 2021) and
carbon dioxide fluxes across the air-sea interface (Olivarez et al. 2022). Here, we focus on the
work of McKinnon and Deser (2018 and 2021) who constructed an Observational LE for global
sea level pressure (SLP) and terrestrial precipitation and temperature based on ~100 years of
monthly gridded instrumental data. To test the skill of their method, they applied it independently
to each member of a climate model LE and then compared the results to the "true" statistical
properties of the model's internal variability based on the full set of ensemble members. According
to this test, their approach was found to be accurate to within 10-20% at most locations. They then
constructed a large (1000 member) ensemble of plausible "parallel worlds" of what the
observational record might have looked like had a different sequence of internal variability
unfolded by chance. Their Observational LE has been used for many applications, including
evaluation of internal variability in climate model LEs, assessment of uncertainty in observed 50-
year climate trends, and quantification of extreme precipitation risk over the Upper Colorado River
basin, a critical water resource for the western US (McKinnon and Deser 2018 and 2021).

***d. Dynamical Adjustment***
Determining the forced contribution to observed changes in climate remains an ongoing challenge.
Most "Detection and Attribution" methods rely on climate models to provide a set of spatial and
temporal "fingerprints" of forced climate change that are distinct from patterns of internal
variability (Hegerl et al. 2007; Santer et al. 2019; Sippel et al. 2019). These model-based
"fingerprints" are then used to assess the proportion of observed climate change that is due to
external forcing. However, model shortcomings may limit the accuracy of such methods. Thus, it
is also desirable to develop complementary approaches to attribution that do not rely on climate
model information. Two such methods, Linear Inverse Modeling (Newman, 2007) and Low-
Frequency Pattern Analysis (Wills et al. 2020), leverage the assumption that forced climate change
evolves slowly compared to the time scales of internal variability. However, decadal shifts in





regional anthropogenic aerosol emissions during the industrial era present challenges to this
assumption and may complicate interpretation of the results (Deser et al. 2020; Persad et al. 2018).

A complementary, physically-based approach to isolating the externally-forced response in
observations without reliance on climate model information is the technique of "Dynamical
Adjustment". This method aims to remove the influence of atmospheric circulation variability
from surface climate anomalies, thereby revealing the thermodynamically-induced component of
observed climate change (Wallace et al. 2013; Smoliak et al. 2015; Deser et al. 2016). According
to the current generation of coupled climate models, the forced component of extra-tropical
atmospheric circulation changes is small relative to internal variability (Deser et al. 2012;
Shepherd, 2014). If models are correct in this regard, then dynamical adjustment can be used to
parse the relative contributions of internal dynamics and forced thermodynamics to observed
climate changes at middle and high latitudes (Wallace et al. 2013; Deser et al. 2016). A variety of
dynamical adjustment algorithms have been developed and tested within the framework of a model
LE (Deser et al., 2016; Lehner et al., 2017 and 2018; Smoliak et al., 2015; Guo et al. 2019;
Merrifield et al., 2017; Terray 2021; Sippel et al. 2019). These protocols are all based on statistical
associations between patterns of SLP and surface climate anomalies deduced from long
observational records. Generally, the data are high-pass filtered or detrended so as to avoid aliasing
any potential forced component onto the statistical relationships. These procedures generally work
best for large-amplitude SLP anomaly patterns, and are more effective for temperature than
precipitation due to higher levels of noise in the latter (Guo et al. 2019).

**2. Data and Methods**

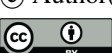



We make use of a state-of-the-art 100-member LE conducted with the National Center for
Atmospheric Research (NCAR) Community Earth System Model version 2 (CESM2), described
in Rodgers et al. (2021). This publicly-available LE resource is unprecedented for its combination
of large ensemble size, high spatial resolution (approximately 1° in both latitude and longitude),
and length of simulation (1850-2100). Each ensemble member is driven by the same radiative
forcing scenario (historical from 1850-2014, and SSP3-7.0 from 2015-2100), but begins from a
different state on 1 January 1850, taken from a long pre-industrial control simulation. We analyze
linear trends over the past 50 years (1972-2021) and projected for the next 50 years (2022-2071).
It should be noted that memory of the initial state is negligible by the middle of the 20th century;
thus, diversity in trends amongst the individual ensemble members is solely due to different
random samples of internal variability, which are superimposed upon a common forced response.

For consistency with the 100-member CESM2 LE, we make use of the first 100 members of the
Observational LE (OBS LE) constructed by McKinnon and Deser (2018) to illustrate the diversity
of past 50-year trends consistent with the statistical spatio-temporal properties of internal
variability in the observational record. For the purpose of comparing directly to the CESM2 LE,
we have added the model's forced trend to the internal trend of each OBS LE member. The OBS
LE is based on the Berkeley Earth Surface Temperature (BEST) dataset (Rohde et al. 2013), the
Global Precipitation Climatology Centre (GPCC) dataset (Schneider et al. 2008), and the
Twentieth Century Reanalysis version 2c (20CR) sea level pressure (SLP) dataset (Compo et al.

219    2011).




We apply the dynamical adjustment methodology of Deser et al. (2016) based on SLP "constructed
circulation analogues" to monthly temperature and precipitation during 1900-2021, using the same
observational data sets as in the OBS LE.  The reader is referred to Deser et al. (2016) for details
of the methodology, and to Lehner et al. (2017 and 2018), Guo et al. (2019) and Terray (2021) for
additional applications.

For each ensemble member of the CESM2 and OBS LEs, we form monthly anomalies by
subtracting the long-term means for each month individually, and then form seasonal averages
(December-February) of the monthly anomalies. We compute 50-year trends of the wintertime
anomalies using linear least-squares regression analysis.  All results shown in this study are
original findings.

**3.  European climate trends**
We begin by illustrating the diversity of winter temperature and precipitation trends over Europe
during the past 50 years (1972-2021) in the CESM2 and OBS LEs (Sections 3a and b), and
projected for the next 50 years (2022-2071) in the CESM2 LE (Section 3c). We then provide a
more quantitative view of the relative contributions of forced climate change and internal
variability to past and future climate trends using a variety of signal-to-noise metrics, with
comparison between the CESM2 and OBS LEs (Section 3d). We summarize the CESM2 LE
results by showing the "expected range" of trend outcomes in Section 3e. Finally, we apply the
technique of "dynamical adjustment" to estimate the forced component of observed temperature
trends (Section 3f), and then use this estimate in conjunction with the OBS LE to produce a purely



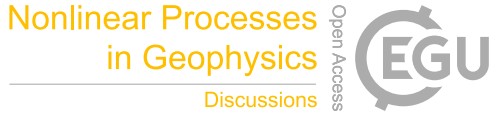

observational estimate of the plausible range of temperature trend outcomes over the past 60 years
(Section 3g).

***a. Past trends (1972-2021) in the CESM2 LE***
The CESM2 model simulates a wide range of wintertime temperature trend patterns for the past
50 years due to the combined effects of internal variability and forced response, as illustrated by
the first 28 members of the LE (Fig. 1). Recall that the only reason that these trend maps are not
identical is because of random differences in internal variability between the members. While
moderate warming is seen over most of the European continent in the majority of cases, as
expected, some members show regions of considerably greater temperature increase (in excess of
1°C per decade for example members 1, 10 and 18), while others exhibit weak cooling in some
locations (for example, members 17, 23 and 26; Fig. 1). The relative contributions of internal
variability and forced response can be readily discerned by comparing the individual member
trends with the ensemble-mean trend (see "EM" panel in Fig. 1). The observed trend ("OBS"
panel in Fig. 1) bears a close resemblance to the model's forced trend in both amplitude and spatial
pattern. This correspondence may be coincidental, as individual members of the CESM2 LE also
resemble the forced response (for example, members 6 and 21), or it may suggest that the model
overestimates the amplitude of internally-generated 50-year trends relative to forced trends. The
OBS LE results shown below will shed some light on these two possibilities.

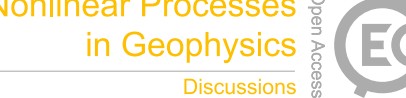

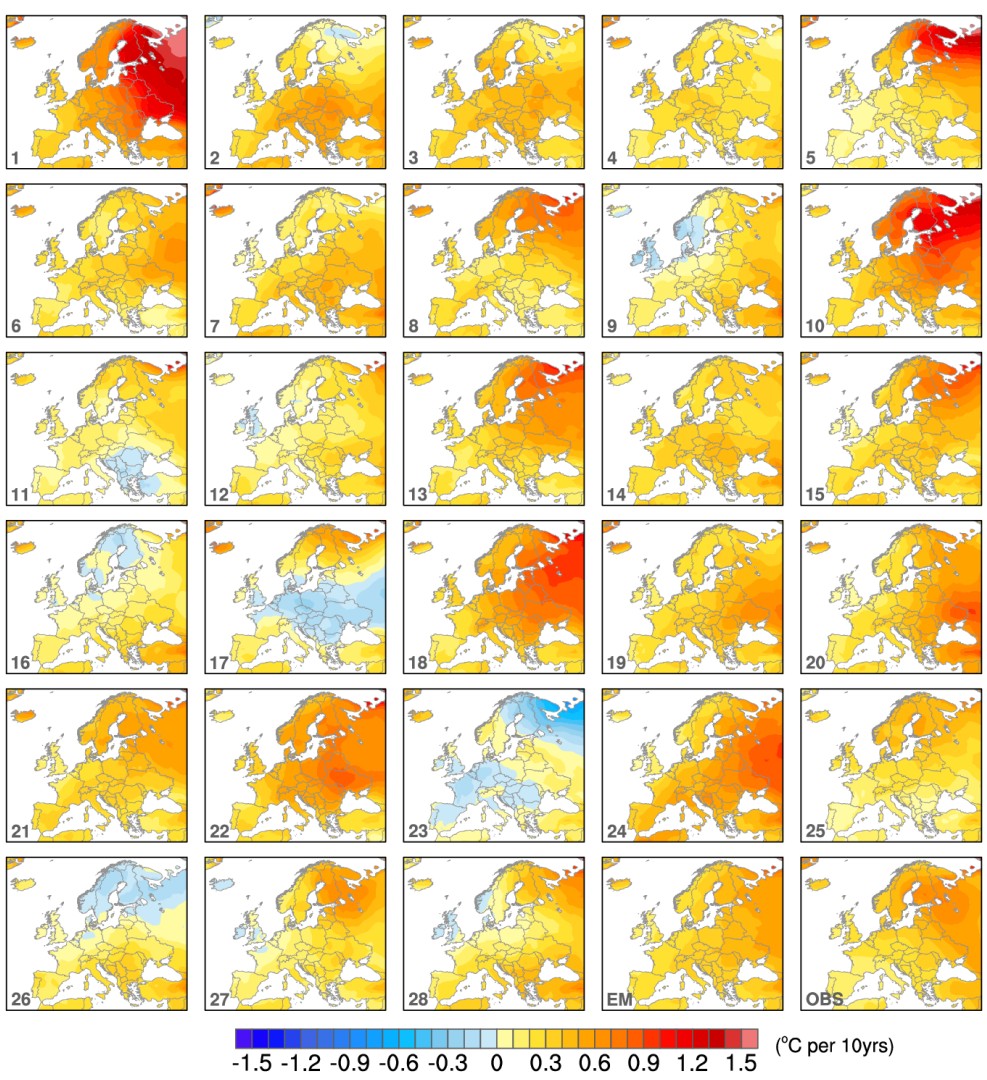

**Figure 1.** Winter air temperature trends (°C per decade) for the period 1972-2021 as simulated by the first 28 members of the CESM2 Large Ensemble (number in the lower left of each panel denotes the ensemble member) and the 100-member ensemble-mean (panel labeled "EM"). Observed trends are shown in the lower right (panel labeled "OBS").



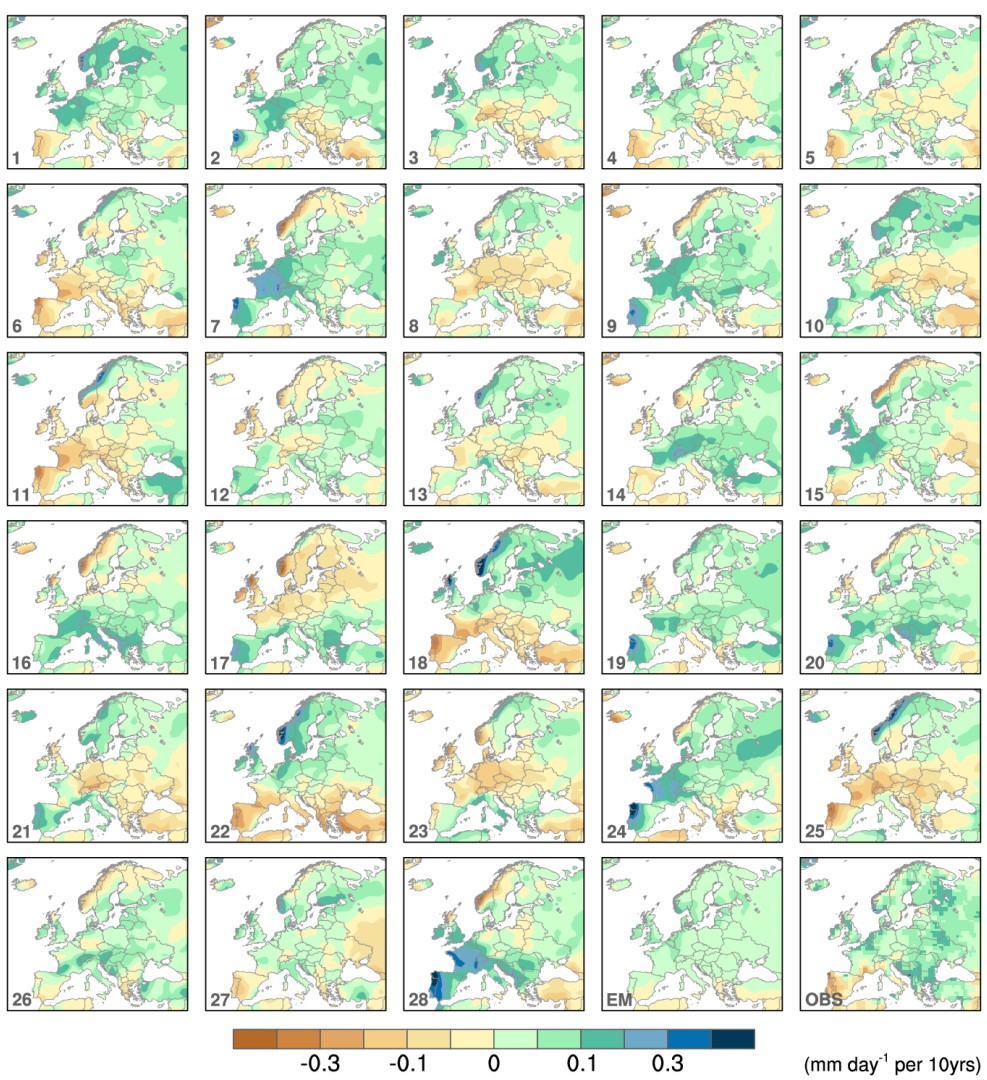

**Figure 2.** As in Fig. 1 but for precipitation (mm d-1 per decade).

Like temperature, precipitation trends also vary considerably across ensemble members (Fig. 2).
While the ensemble-mean trend shows modest increases in precipitation throughout Europe
(except for the southernmost fringes), internal variability can evidently overwhelm the forced
response in individual simulations. For example, some members show drying over large parts of





the continent, while others depict enhanced wetting in the same regions (compare, for example,
members 22 and 28, which show nearly opposite patterns). Observed precipitation trends are
generally positive, except over Spain, Portugal, southern France and other parts of the western
Mediterranean (Fig. 2). The observed precipitation increases, while of the same sign as the
model's forced response, are approximately twice as large in many areas. Again, the interpretation
of the observed trends is ambiguous, since there are individual members that resemble
observations (for example, member 1).

***b. Past trends (1972-2021) in the OBS LE***
The individual members of the OBS LE show a qualitatively similar diversity of 50-year
temperature trends as the CESM2 LE (Fig. 3). Like CESM2, some members show weak cooling
in some areas while others show widespread moderate or strong warming. This suggests that the
resemblance between the observed trend and the model's forced response may be purely
coincidental. Precipitation trends in the OBS LE also display large contrasts between members,
similar to CESM2 (Fig. 4). For example, nearly opposite patterns are found between members 6
and 11 (or 8 and 9). Trend amplitudes also vary considerably across the OBS LE, with larger
magnitudes in some members (for example, members 3 and 20) compared to others (e.g., members
21 and 13). While no single member of the 28 OBS LE samples shown matches the model's forced
trend, member 21 with its relatively muted trends comes close.





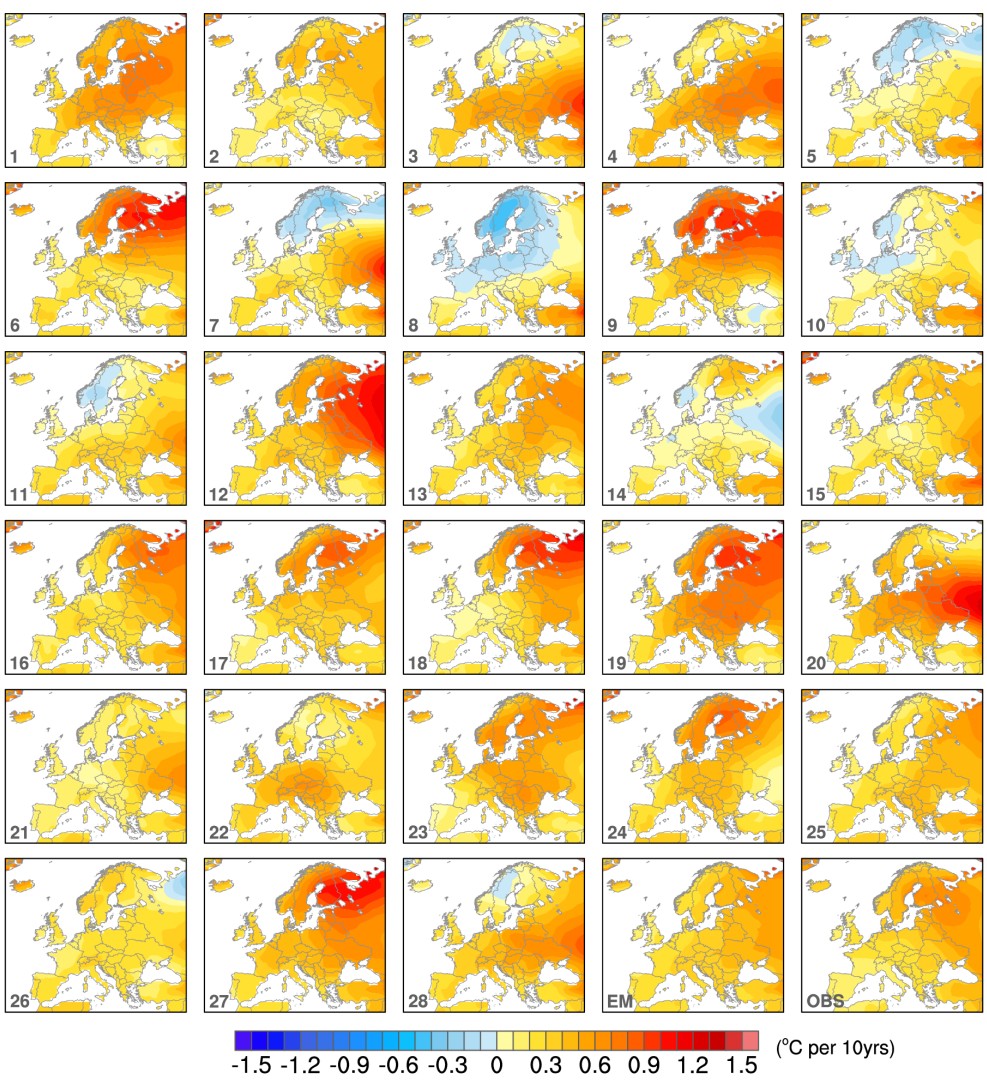

**Figure 3.** As in Fig. 1, but for the Observational Large Ensemble of McKinnon and Deser (2018) with the ensemble-mean from the 100-member CESM2 Large Ensemble. See text for details.


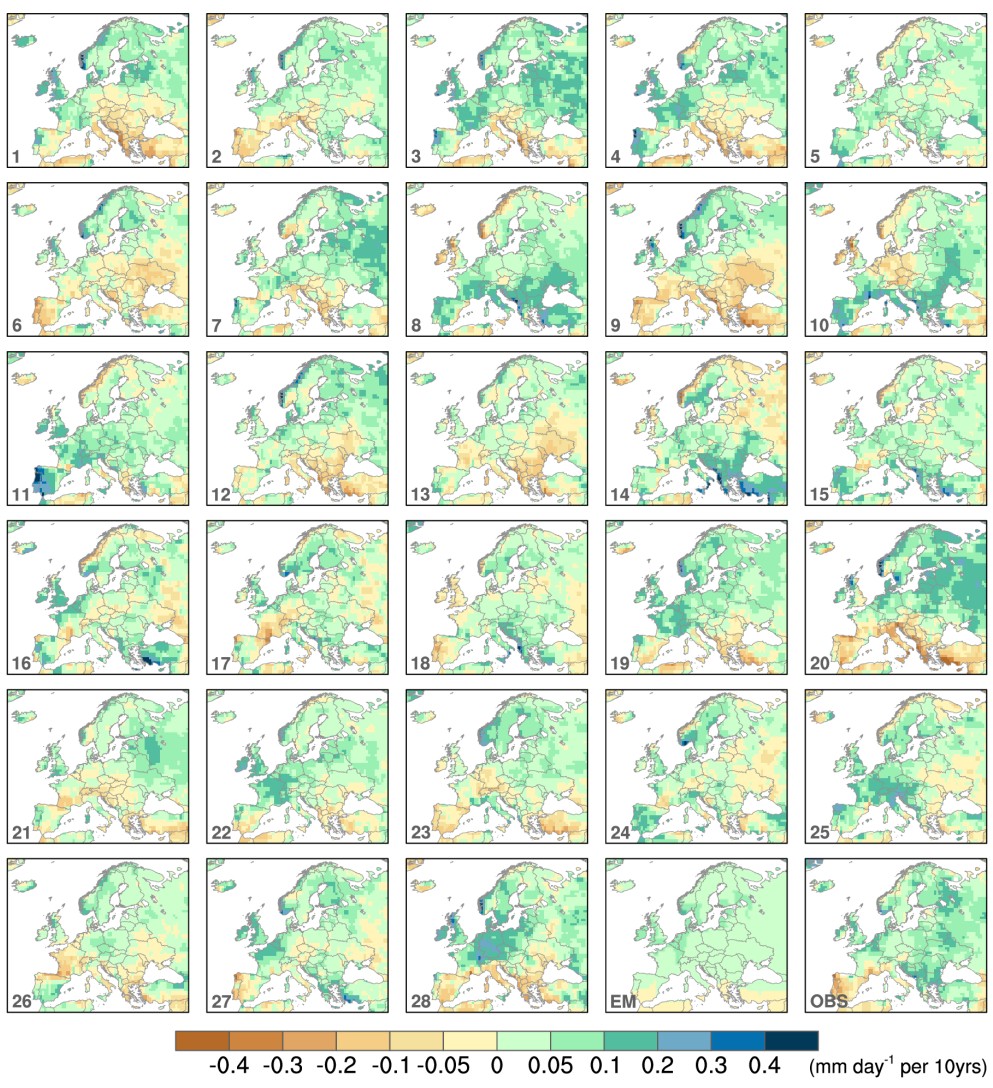

Figure 4. As in Fig. 2, but for the Observational Large Ensemble of McKinnon and Deser (2018) with the ensemble-mean from the 100-member CESM2 Large Ensemble. See text for details.



*c. Future Trends (2022-2071) in the CESM2 LE*

As expected, temperature trends projected for the next 50 years show larger amplitudes than those for the past 50 years in the CESM2 LE (Fig. 5). This is due to the fact that the forced (ensemble-mean) component of warming increases as greenhouse gas emissions accelerate. In most regions, the forced warming trend increases by approximately 0.2°C per decade in the future compared to the past. Notable exceptions are Iceland and the British Isles, which show less warming in the future due to a circulation-induced forced cooling trend (see Section 3e). Despite a larger forced component, temperature trends projected for the next 50 years still show a wide range of amplitudes across individual members of the CESM2 LE. For example, member 13 is striking for its muted warming (generally < 0.5°C per decade) across Europe (and absolute cooling over the UK and Iceland), while member 28 shows highly amplified warming, with values exceeding 1.3 °C per decade over western Russia.

Forced trends in precipitation are projected to amplify over the next 50 years, with greater wetting over northern Europe and drying over southern Europe and the Mediterranean (Fig. 6). In addition, the region with a forced drying trend is projected to expand northward into Spain, Italy and the Balkan Republics. While the forced pattern of future drying in the south and wetting in the north is generally evident in most of the simulations shown, there are notable differences in amplitude across the members. For example, member 28 shows precipitation trends in excess of 0.1 mm $d^{-1}$ per decade over most of northern Europe, while member 11 shows positive precipitation trends of less than half this amount. Members 27 and 28 illustrate that the mid-section of the European continent may get wetter or drier depending on the unpredictable sequence of internal variability



that unfolds. Thus, internal variability can still make a sizeable contribution to the projected
patterns and amplitudes of winter precipitation trends over the next 50 years.

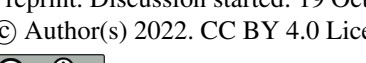

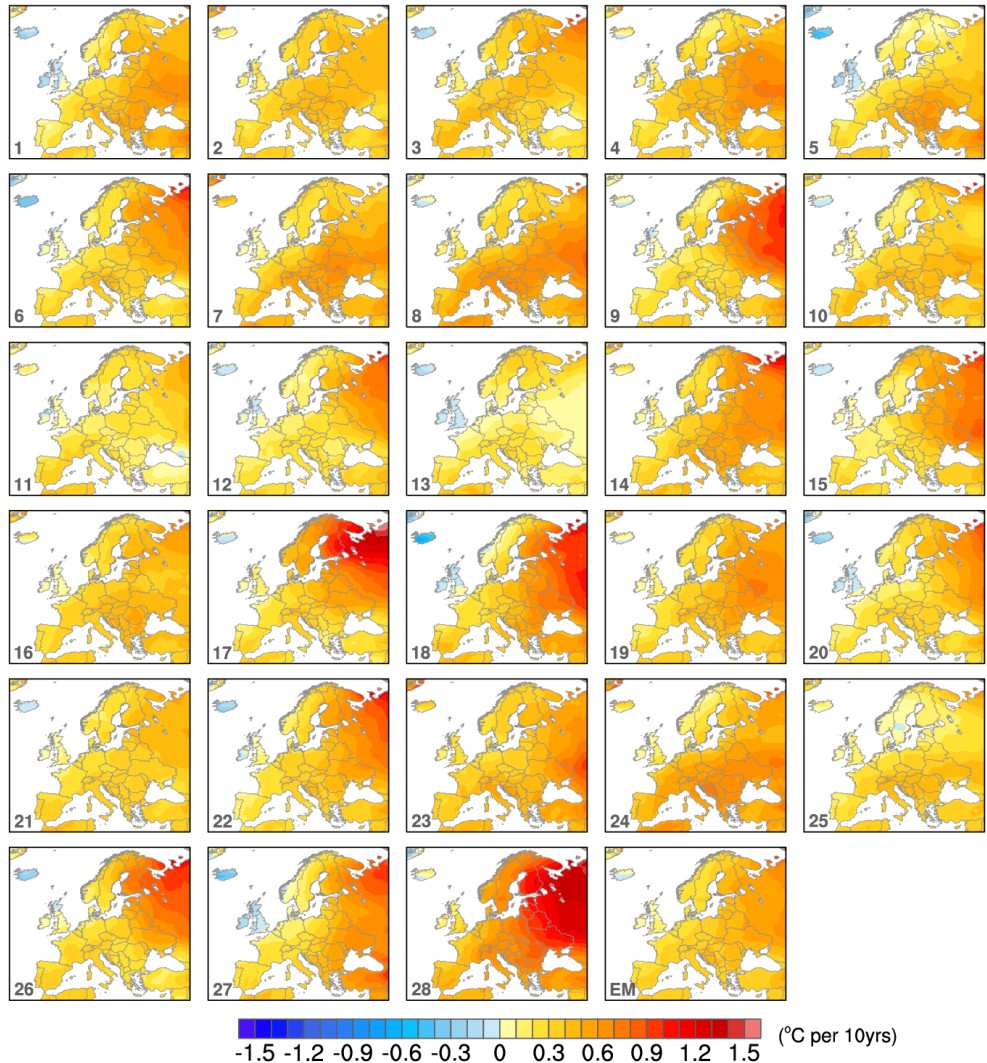

**Figure 5**. As in Fig. 1, but for the period 2022-2071.





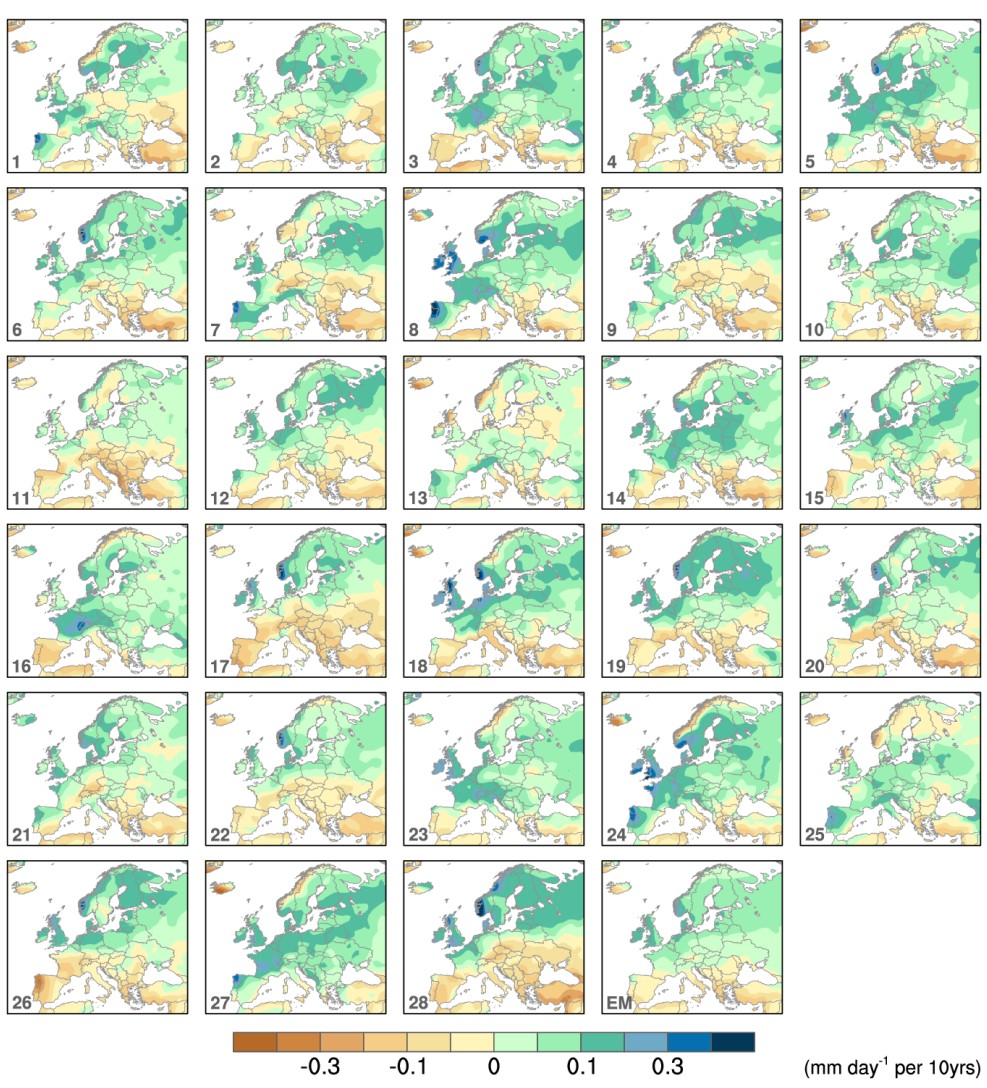

**Figure 6**. As in Fig. 2, but for the period 2022-2071.





***d. Signal-to-noise metrics and model evaluation.***
In the previous section, we conveyed a qualitative impression of the possible range of 50-year
trends due to the superposition of internal variability and forced climate change in the CESM2 and
OBS LEs. Here, we provide a more quantitative view, beginning with a comparison of the standard
deviation ($\sigma$) of trends over the period 1972-2021 computed across the ensemble members of each
LE.  In the CESM2 LE, the ensemble $\sigma$ of temperature trends increases from southwest to
northeast, with minimum values (0.05-0.10 K per decade) over Spain and northern Africa, and
maximum values (0.30-0.35 0.5°C per decade) over northwestern Russia (Fig. 7a).  A similar
pattern is found in OBS LE, with some regional differences in amplitude (Fig. 7b).  In particular,
the ensemble $\sigma$ values are significantly smaller (20-40%) over Scandinavia, Germany and Poland,
and significantly larger (20-40%) in areas near the Mediterranean and Black Seas, in the OBS LE
compared to the CESM2 LE (Fig. 7c).  For precipitation trends, the two LEs show similar patterns
of ensemble $\sigma$, with largest amplitudes generally along the west coasts (0.10 - 0.25 mm d$^{-1}$ per
decade) and over southwestern Europe (values 0.05 – 0.10 mm d$^{-1}$ per decade: Figs. 7d and e).
However, CESM2 LE significantly underestimates the OBS LE by more than 40% along the
Mediterranean and Black Seas and parts of Russia, and significantly overestimates the OBS LE by
20-40% in many areas of western Europe (Fig. 7f).


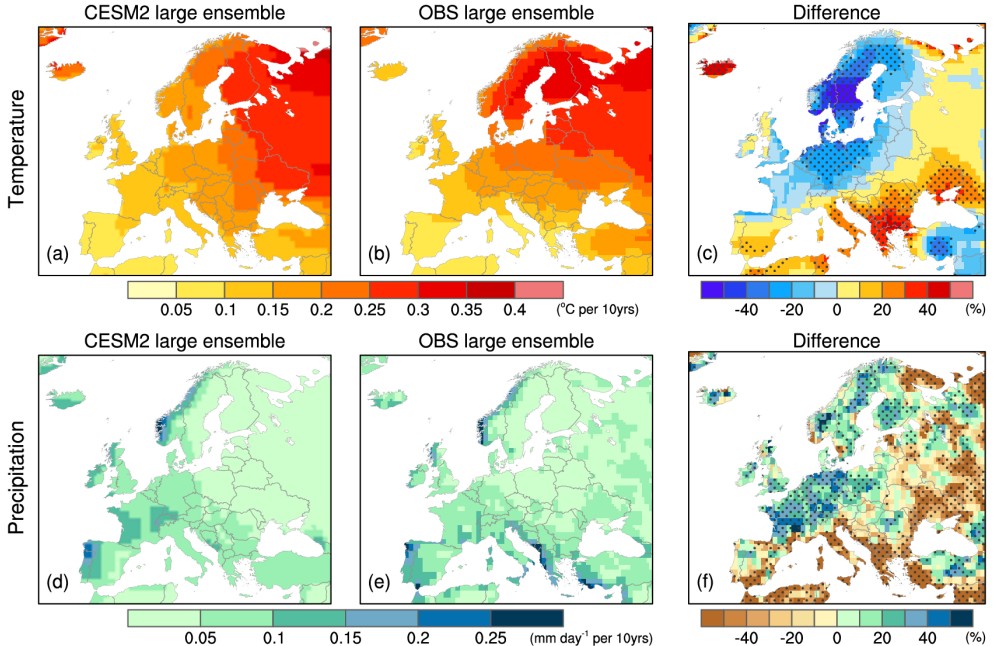

**Figure 7**. Standard deviation of 50-year trends (1972-2021) across 100 members of the CESM2
Large Ensemble (a,d) and 100 members of the Observational Large Ensemble (b,e), and their
difference (c,f) for winter air temperature (top; °C per decade) and precipitation (bottom; mm d$^{-1}$
per decade). Stippling in panels c and f indicates that the differences are statistically significantly
at the 95% confidence level.

Next, we assess the relative magnitude of the forced and internal components of trends by

computing a "signal-to-noise" ratio defined as the CESM2 ensemble-mean trend divided by the σ

of trends across the 100 members of each LE. This "signal-to-noise" ratio indicates whether the

forced trend can be readily detected in any single ensemble member (and by extension, the real

world), or whether internal variability dominates. For a normal distribution, a signal-to-noise ratio

greater than two indicates that the ensemble-mean (forced) trend is significantly different from

zero at the 95% confidence level: that is, there is less than a 5% chance that the ensemble-mean

trend could have been a result of random internal variability. In CESM2, the signal-to-noise of

forced temperature trends over the past 50 years generally ranges from 1.5 - 2 over central and


northern Europe, and from 2-3 over southern Europe (Fig. 8a). Forced precipitation trends over
the past 50 years exhibit much lower signal-to-noise ratios than temperature, with values generally
< 1 and nearly always < 1.5 (Fig. 8d).

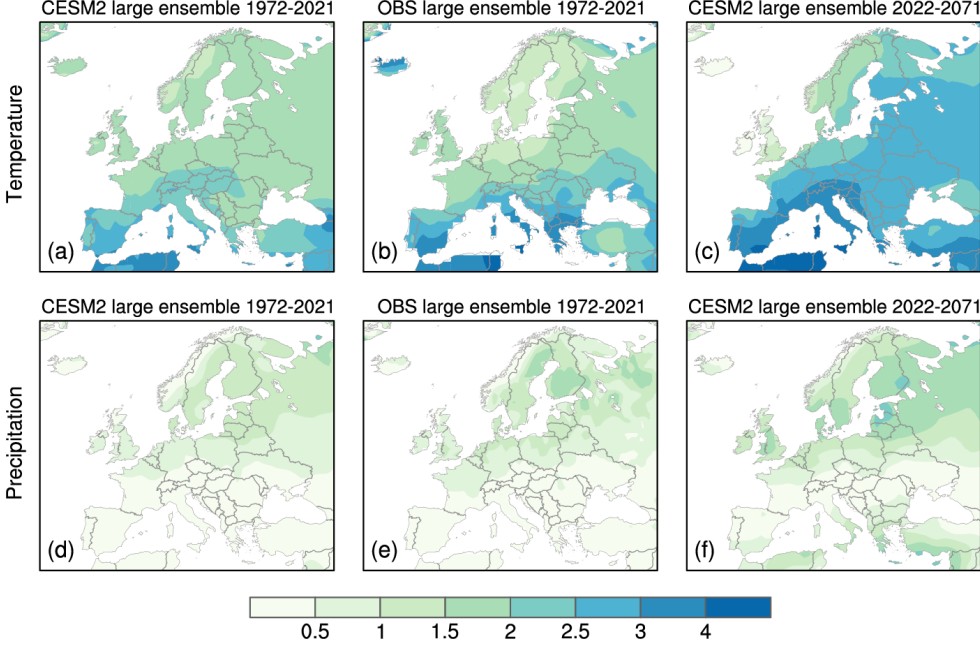


**Figure 8**. Signal-to-noise of forced trends in winter (top) air temperature and (bottom) precipitation based on the 100-member CESM2 Large Ensemble during 1972-2021 (a,d), the Observational Large Ensemble during 1972-2021 (b,e), and the CESM2 Large Ensemble during 2022-2071 (c,f). See text for details.


How much do model biases in ensemble σ shown previously affect the signal-to-noise of the
model's forced trends? We address this question by using the OBS LE σ values in place of the
model's σ values in the signal-to-noise calculation. This substitution results in an enhancement of
signal-to-noise of past forced temperature trends over southern Europe and a reduction in signal-
to-noise over Scandinavia, Germany and Poland, with a net increase from 38% to 60% in the area
with values > 2 (Fig. 8b). The impact of model biases in ensemble trend σ is much less pronounced





for precipitation than temperature, with signal-to-noise values in all locations remaining below 2
(Fig. 8e).

As expected, signal-to-noise values are higher for forced trends in the future than in the past.
Ninety-seven percent of the area of the continent (excluding Iceland and Greenland) shows a
signal-to-noise value > 2 for forced temperature trends during 2022-2071 (Fig. 8c), compared with
38% for trends during 1972-2021. Forced precipitation trends in the future remain uncertain, with
only 2% of the land area showing a signal-to-noise value > 2 (Fig. 8f).

Another way to view the relative impacts of internal variability and external forcing on trends is
by computing the fraction of ensemble members at each location that show a positive trend (e.g.,
warming or wetting). This metric conveys the likelihood of having a positive (or negative) trend
in any single ensemble member, which is analogous to the single "realization" of the real world.
At nearly all locations, more than 95% of ensemble members in the CESM2 LE show warming in
both the past and future periods, with slightly lower percentages (85-95%) over western
Scandinavia and parts of Great Britain (and < 75% over Ireland, Scotland and Iceland in the
future); (Figs. 9a and c).  Similar percentages are obtained when the internal component of past
temperature trends in the OBS LE is used in place of the model's internal trends, with some
reduction (75-95%) over Scandinavia, northern Russia, Germany and Poland (Fig. 9b).


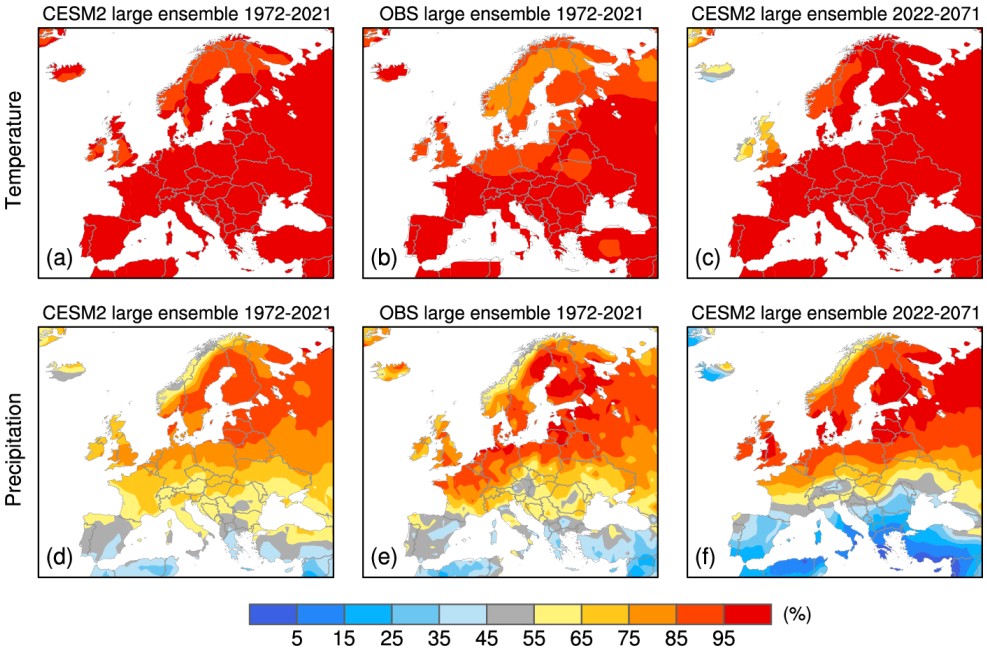

**Figure 9**. The percentage of ensemble members with a positive trend in winter (top) air temperature and (bottom) precipitation trends based on (a,d) the 100-member CESM2 Large Ensemble during 1972-2021, (b,e) the 100-member Observational Large Ensemble during 1972-2021, and (c,f) the 100-member CESM2 Large Ensemble during 2022-2071.

The sign of the trend in any given ensemble member is more uncertain for precipitation than for temperature. The highest chances (> 85%) of a positive precipitation trend are found over the northernmost third of the continent excluding Norway, both in the past and future (Figs. 9d and f). Similarly high chances of a negative precipitation trend (equivalent to < 15% of a positive trend) occur in areas near the Mediterranean Sea, but only in the future. The central portion of the continent shows roughly equal chances of having a positive or negative trend, both in the past and future. The area with a > 85% chance of a positive precipitation trend in the past 50 years expands southward into northern France, Germany and areas bordering the Baltic Sea when internal variability is derived from the OBS LE compared to the CESM2 LE (Fig. 9e).





Taken together, the results shown in Fig. 9 indicate that warming is virtually guaranteed at nearly
all locations, both in the past 50 years and the next 50 years, according to the CESM2 LE.
However, the sign of the precipitation trend (past and future) is robust only over the northern tier
of the continent, and only in the future over the Mediterranean region. The model results for past
trends are found to be generally credible as measured against the OBS LE, with some
overestimation in north-central Europe.

***e. Range of outcomes and the role of the atmospheric circulation***
As the saying goes, "climate is what we expect, weather is what we get". This adage is also
applicable to climate change, where "human-induced climate change is what we expect, internal
variability plus human-induced climate change is what we get" (Deser 2020). Here, we illustrate
"what we expect" and the range of "what we get" for past and future 50-year trends in the CESM2
LE, using the ensemble-mean for "what we expect" and two contrasting ensemble members for
the range of "what we get". We select the contrasting members from the bottom and top 5$^{th}$
percentiles of the distribution of 100-member trends averaged over the European continent for
each period separately. This selection criterion is somewhat arbitrary and does not necessarily
capture the wide range of trend amplitudes that may occur at a single location or sub-region, nor
does it portray the full range of spatial patterns that occur within the ensemble.

There is a large range in temperature trend outcomes ("what we get") for both the past 50 years
and the next 50 years as depicted by the "warm" and "cool" end-members (Fig. 10). For past
trends, the "warm" end-member shows temperature increases of 0.9-1.1 °C per decade over the
eastern portion of the continent (Fig. 10b), while the "cool" end-member displays muted warming

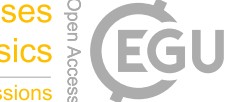

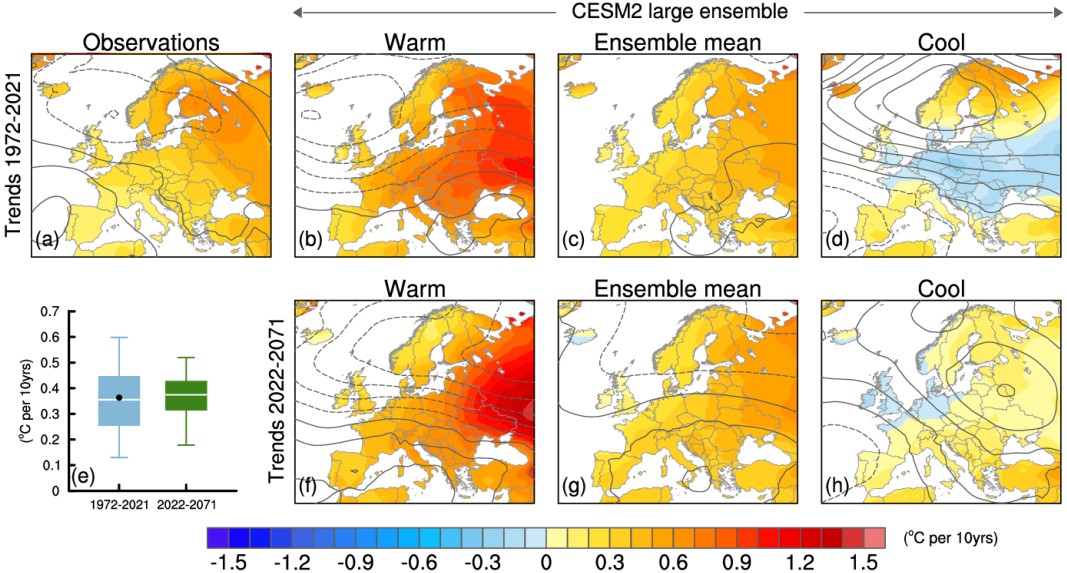

**Figure 10. A Range of Outcomes.** Trends in winter air temperature (color shading; °C per decade) and sea-level pressure (SLP) (contours; contour interval of 0.25 hPa per decade, negative values dashed) for the period (top) 1972-2021 and (bottom) 2022-2071. Panel (a) shows observed trends (1972-2021) and remaining panels show simulated trends from the 100-member CESM2 Large Ensemble: (c,g) ensemble-mean; (b,f) "warm" end-member; (d,h): "cool" end-member. See text for details. Panel (e): Distribution of European-average trends for 1972-2021 (blue) and 2022-2071 (green) from the CESM2 Large Ensemble (box outlines 25th-to-75th percentile range, whiskers mark the 5th-to-95th percentile range, the horizontal white line denotes the median value, and the black circle marks the observed value).

(< 0.3 °C per decade) and even slight cooling through the midsection of the continent (Fig. 10d).

Clearly, the forced trend ("what we expect"), which depicts moderate warming (0.2-0.6°C per

decade) across the continent does not tell the whole story (Fig. 10c). Analogous results are found

for trends projected over the next 50 years: the "warm" member shows temperature increases of

1.0-1.5 °C per decade over west-central Russia (Fig. 10f) while the "cool" member depicts < 0.2°C

per decade warming over most of the continent (Fig. 10h), in marked contrast to the forced trend

which ranges from 0.3-0.6°C per decade (Fig. 10g). As discussed previously, the observed

temperature trend map resembles the model's ensemble-mean, but this could be by chance (Fig.

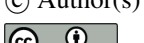



10a). In terms of European averages, the observed trend (0.36 °C per decade) is nearly coincident
with the median value of the model's trend distribution, which has a 5th-to-95th percentile range of
0.13-0.60 °C per decade for past 50-year trends (Fig. 10e). Curiously, the model's median trend
value for Europe as a whole increases only slightly in the future compared to the past, while the
5th-to-95th percentile range narrows slightly (Fig. 10e). Further work is needed to understand why
this is the case.

As mentioned in Section 1d, previous work has shown that internal variability of the large-scale
atmospheric circulation causes much of the member-to-member differences in temperature trends
in model LEs. Here, we provide a qualitative indication of the circulation influence by
superimposing SLP trends upon the maps in Fig. 10. In the case of past trends, the "warm" member
shows a positive North Atlantic Oscillation (NAO)-like pattern (Hurrell et al. 2003), with negative
SLP trends centered near Iceland and positive SLP trends centered over the Mediterranean (Fig.
10b). This SLP pattern is indicative of stronger westerly/southwesterly flow, which brings
relatively warm maritime air over the continent. The "cool" member shows a largely opposite
flow configuration (albeit with longitudinal shifts in the SLP centers-of-action), which advects
relatively cold air from the east over the continent (Fig. 10d). In comparison, the forced response
shows negligible atmospheric circulation change (Fig. 10c). Striking contrasts in circulation are
also found for the future period, with a large positive NAO-like trend pattern in the "warm"
member and a blocking continental "High" in the "cool" member (Figs. 10f and h). Future trends
in SLP also contain a modest forced component indicative of enhanced westerlies over the
continent (Fig. 10g).





The "wet" and "dry" end-members also show striking regional contrasts in both precipitation and
circulation (Fig. 11). For example, for past trends, the "wet" member shows precipitation increases
of 0.2-0.3 mm d$^{-1}$ per decade over France, southern Germany, Portugal and the UK, and
precipitation declines over northern Norway and along the Mediterranean Sea (Fig. 11b).  A nearly
opposite pattern is found for the "dry" member (Fig. 11d). These contrasting precipitation trends
can be understood in the context of the overlying atmospheric circulation changes, with wetter
areas coinciding with anomalous westerly/southwesterly flow and drier areas located under
blocking anticyclones.  Analogous patterns are found for future trends, with pronounced increases
in precipitation over western Europe associated with the low pressure trend centered over the
British Isles in the "wet" member (Fig. 11f), and generally reduced precipitation in the "dry"
member associated with the blocking High centered over southern Europe (Fig. 11h).

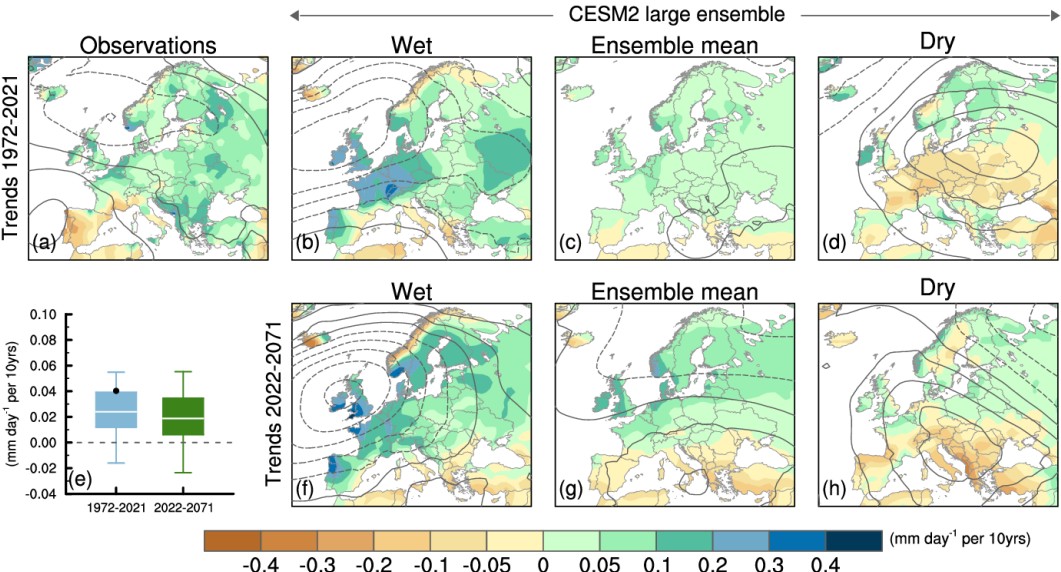

**Figure 11**.  As in Fig. 10 but for precipitation (mm d$^{-1}$ per decade).




*f. Unmasking forced climate change in observations via "Dynamical Adjustment"*

The empirical method of "dynamical adjustment" introduced in Section 1d can be used to estimate
the circulation-induced component of observed temperature anomalies; this dynamically-induced
contribution can then be subtracted from the original anomaly to obtain the thermodynamically-
induced component as a residual. Since this method uses no information from climate models, it
provides an independent estimate of the thermodynamic component of observed temperature
trends, which can be compared with the forced response simulated by climate model LEs.

Figure 12 shows the decomposition of observed DJF temperature trends into their dynamical and
residual thermodynamic contributions. For this example, we have used the 60-year period 1962-
2021 when observed SLP trends are more than twice as large as those during 1972-2021 on a per
decade basis (compare SLP contours in Figs. 10a and 12a). Observed SLP trends during the past
60 years show a pronounced positive NAO-like pattern, with maximum negative values of -1.25
hPa per decade near Iceland and maximum positive values of +0.75 hPa per decade west of Spain
(Fig. 12a). Enhanced westerly/southwesterly flow associated with this pattern advects warm air,
raising surface temperatures by 0.1- 0.3°C per decade (with maximum warming over northern
Europe) according to the dynamical adjustment algorithm (Fig. 12b). Removing this dynamically-
induced component from the total trend reveals the residual thermodynamic contribution to the
observed warming trend (Fig. 12c). This observed thermodynamic trend is much closer in
amplitude (and arguably pattern) to the model's forced response, given by the CESM2 LE
ensemble-mean trend (Fig. 12d), than is the total observed trend. Further, the lack of an
appreciable forced SLP trend in CESM2 indicates that the model's forced temperature trend is





thermodynamically-driven. The level of agreement between the observed thermodynamic
temperature trend and the model's forced thermodynamic trend leads to two powerful conclusions:
1) the model's forced temperature trend is realistic; and 2) removing the circulation-induced
component from the observed trends can effectively reveal the influence of anthropogenic forcing.
Analogous results have been found for North America (Deser et al. 2016).

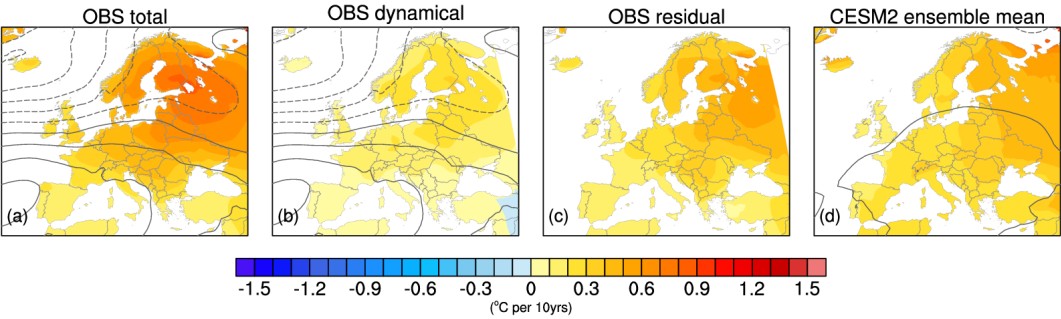


**Figure 12**. Decomposition of (a) observed winter air temperature trends (1962-2021; °C per
decade) into (b) dynamical and (c) residual thermodynamic contributions using the "dynamical
adjustment" procedure of Deser et al. (2018) based on constructed circulation analogues (see text
for details). Contours in (a) show observed sea-level pressure (SLP) trends (contour interval of
0.25 hPa per decade, negative values dashed); contours in (b) show the observed SLP trends
estimated from the constructed circulation analogues; contours in (c) based on the difference
between (a) and (b) are near-zero and not shown. Panel (d) shows the ensemble-mean temperature
and SLP trends from the 100-member CESM2 Large Ensemble (note that only the zero contour
shows up in panel d).

Precipitation is an inherently noisier field than temperature in both time and space, making it
challenging to extract the forced signal via "dynamical adjustment"; indeed, only one previous
study has attempted dynamical adjustment of observed precipitation trends (Guo et al. 2019).
Keeping in mind that the estimate of the circulation-induced component of precipitation trends
may be less robust than for temperature, we present the results as a proof-of-concept. Observed
precipitation trends during 1962-2021 are mainly driven by changes in atmospheric circulation,
with a small thermodynamic residual component (Fig. 13). This residual component bears some

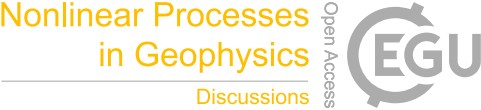

resemblance to the forced response in CESM2, particularly in terms of amplitude (~ 0.05 mm d-1
per decade; Fig. 13d).  Notable areas of agreement in the sign of the trends include drying over
much of southern Europe and wetting over parts of northern Europe; central Europe shows less
agreement in polarity, unsurprisingly since this region was found to have lower signal-to-noise
than other areas.

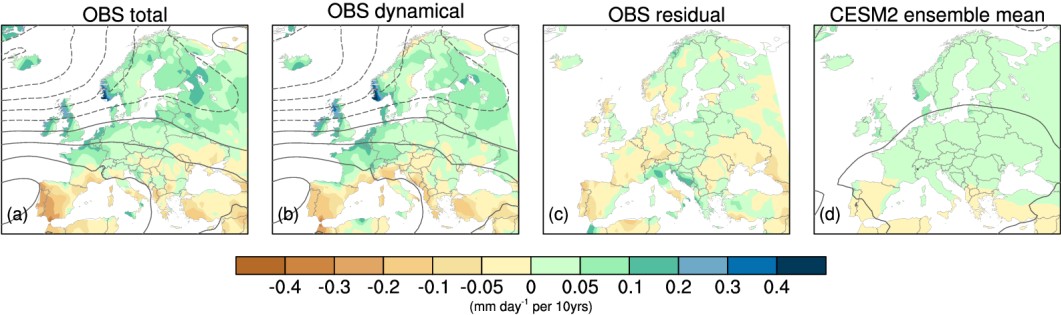


**Figure 13**. As in Fig. 12 but for precipitation (mm d⁻¹ per decade).

***g. Toward an observationally-based "range of outcomes"***
We conclude by bringing together the results of the Observational LE and "dynamical adjustment"
to produce a fully observationally-based estimate of the range of past 60-year trends in temperature
and precipitation.  To the best of our knowledge, this is first time that these two approaches have
been combined. Specifically, we add the internal component of trends from each member of the
OBS LE to the thermodynamic-residual trend (the estimated observed forced response) obtained
from dynamical adjustment. As before, we select two contrasting ensemble members from the tails
of the distribution based on European-wide averages to illustrate the range of trend outcomes. The
"warm" end-member shows pronounced temperature increases over the northern two-thirds of the
continent, with maximum values in excess of 0.9 °C per decade, while the "cool" end-member
warms less than 0.2 °C per decade in most areas and even cools slightly over Ukraine and



neighboring countries (Figs. 14 b and d, respectively). These divergent temperature trends are
associated with contrasting SLP trends, with a positive NAO-like pattern in the "warm" member
a negative (and eastward-shifted) NAO pattern in the "cool" member (Figs. 14 b and d).
Qualitatively, this range of trend outcomes for both temperature and SLP is remarkably similar to
that obtained directly from the CESM2 LE, with some regional differences in the location of
cooling in the "cool" end-member (Figs. 14 e and g). There is no guarantee that the patterns and
amplitudes of trends sampled in our selected end-members will agree between the model and
observationally-based results, since there are many configurations that produce extremes in
European-wide averages (not shown). That there is a strong qualitative resemblance between them
is a testament to both the realism of the model's forced response and internal variability, and the
efficacy of the OBS LE and dynamical adjustment approaches.

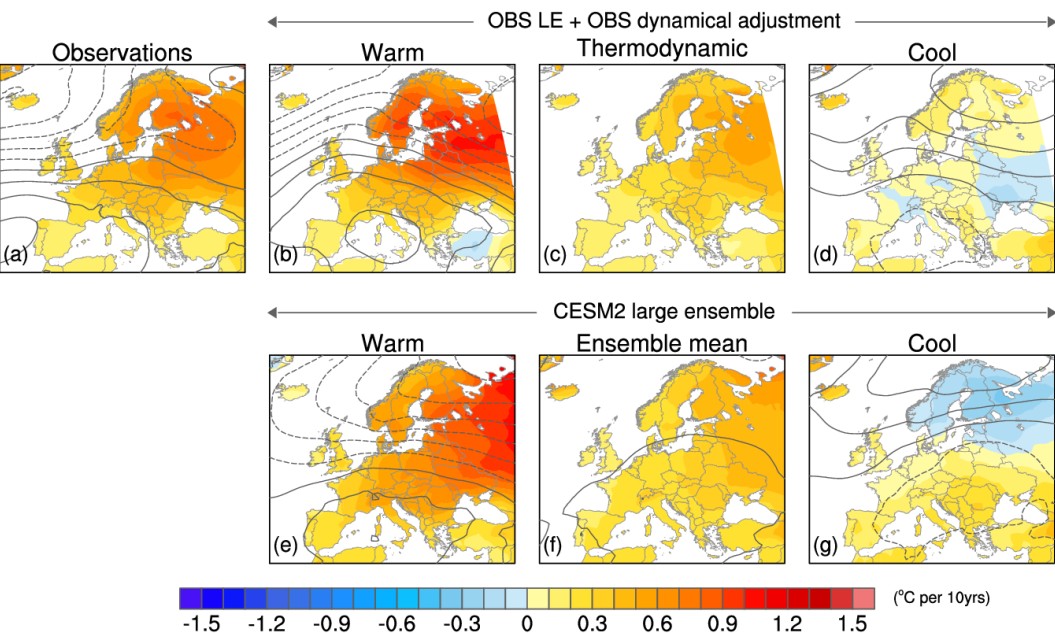


**Figure 14**. As in Fig. 10 but for the period 1962-2021. The top row is based on the Observational
Large Ensemble combined with the residual thermodynamic component of observed trends. The
bottom row is based on the 100-member CESM2 Large Ensemble. See text for details.



Precipitation trends in the "wet" and "dry" end-members are also similar between the model and
observationally-based results (Fig. 15). The "wet" members show widespread increases in
precipitation over southern and central Europe (maximum values of 0.2-0.4 mm d$^{-1}$ per decade)
and drying over the northern UK and parts of Scandinavia (Figs. 15 b and e). Largely opposite
patterns prevail in the "dry" members (Figs. 15 d and g). The contrasting precipitation trends in
the "wet" and "dry" end-members are associated with opposite flow configurations, with regions
of drying corresponding to high pressure and vice versa.

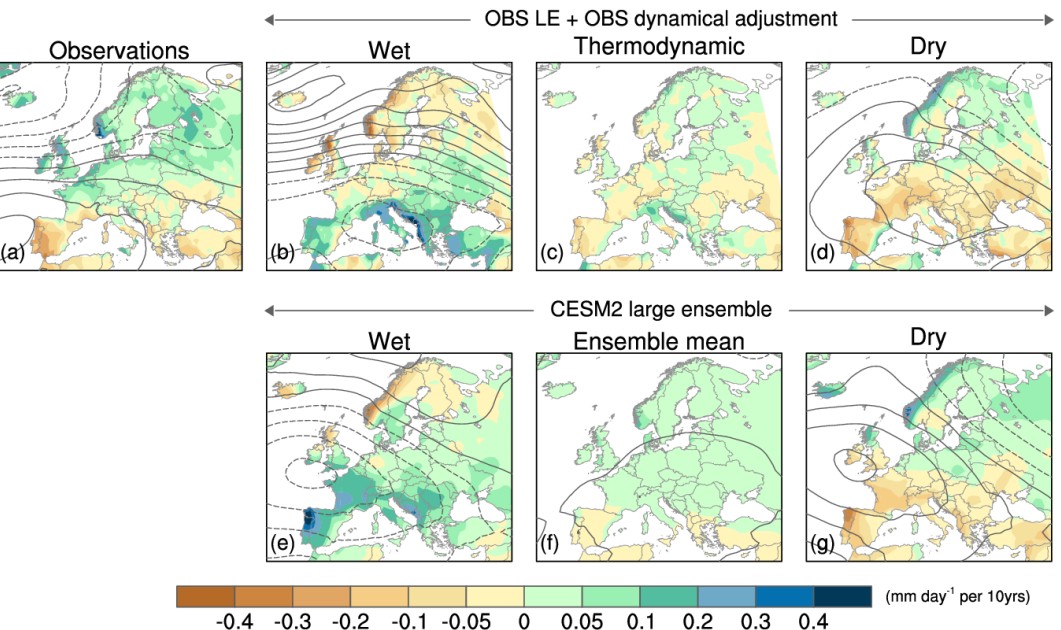


**Figure 15**. As in Fig. 14 but for precipitation (mm d$^{-1}$ per decade).

**4. Summary and open questions**
Disentangling the effects of internal variability and anthropogenic forcing on regional climate
change remains a long-standing issue in climate sciences. Recent advances in climate modeling
and physical understanding have led to new insights on this topic, and provided an improved source



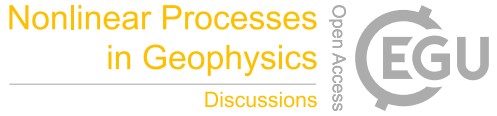

of information on the future risks of climate and weather extremes associated with human-induced
climate change. Here, we have highlighted new findings for European winter climate based on the
following complementary tools: Earth System Model Large Ensemble simulations; an
observationally-based Large Ensemble; and an empirical approach for removing the influence of
internal atmospheric circulation variability from observed climate anomalies.

The new 100-member CESM2 Large Ensemble shows that internal climate variability imparts
considerably uncertainty to past and future 50-year trends in winter temperature and precipitation
over Europe. Such uncertainty is irreducible due to the lack of predictability of the simulated
internal variability on decadal time scales. A novel synthetic Large Ensemble constructed from the
statistical characteristics of internal variability in the observational record exhibits quantitatively
similar levels of uncertainty in past 50-year trends as the CESM2 LE, reinforcing the credibility
of the model's internally-generated trends. Additionally, the results of our "dynamical adjustment"
procedure applied to observations shows good agreement between the observed thermodynamic-
residual trend component and the model's forced thermodynamic trend, further underscoring the
realism of CESM2. Finally, for the first time, we have combined internal variability of trends from
an Observational Large Ensemble with an observational estimate of the forced trend (the
thermodynamic-residual component obtained from "dynamical adjustment") to show what the
observed range of past trends in European temperature and precipitation could have been. Because
it does not rely on climate model information, this observationally-based range of trend outcomes
provides a powerful test of the range of simulated trends in a model Large Ensemble.

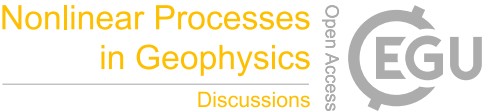

Many outstanding questions remain regarding the relative influences of internal climate variability
and anthropogenic forcing on regional climate change in models and the real world. Fortunately,
promising new tools are being developed to help address these challenges. For example, innovative
machine learning methods may be able to improve upon existing techniques for constructing
Observational Large Ensembles. Such methods have shown good results as statistical emulators
of model-based LEs, but their application to the observational record remains to be pursued
(Beusch et al. 2019). Similarly, neural network approaches to dynamical adjustment may offer
increased skill compared to conventional methods (Davenport and Diffenbaugh, 2021), but have
yet to be applied with the aim of separating forced and internal components of observed trends.
Complementary physically-based approaches such as Linear Inverse Modeling and Low-
Frequency Pattern Analysis mentioned in Section 1d also offer promise for estimating the forced
response in observations without reliance on climate models and should be pursued more widely.

We have relied on the fact that the CESM2 LE (like other models of its class; see Deser et al. 2020
and references therein) simulates a negligible forced atmospheric circulation trend over the past
50-60 years to interpret our observed dynamical adjustment results (i.e, we have equated the
observed dynamically-induced trend with the internal component, and the observed
thermodynamic-residual trend with the forced component). If the model is erroneous in this regard,
then our interpretation of our decomposition of observed trends into "internal dynamical" and
"forced thermodynamic" components is flawed. Indeed, recent work suggests that climate models
may be less predictable on seasonal-to-decadal timescales than the real world, particularly in terms
of the large-scale extra-tropical atmospheric circulation (the so-called "signal-to-noise" paradox;
e.g., Scaife et al. 2014; Eade et al. 2014; Scaife and Smith, 2018). But whether the results from





such initial-value predictability studies carry over to models' forced atmospheric circulation
responses to anthropogenic emissions remains an open question. Finally, a recent study by
Strommen et al. (2002) finds that inclusion of stochastic parameterizations amplifies the simulated
atmospheric circulation response to sea surface temperature and Arctic sea ice anomalies. Such
stochastic parameterizations may represent unresolved air-sea coupling processes in "coarse-
resolution" climate models such as CESM2. Emerging efforts to develop mesoscale-eddy-
resolving global coupled climate models may provide more definitive answers to this elusive
challenge in the near future.

**Data and code availability statement**
All data used in this study are publicly available as follows:
CESM2 Large Ensemble: https://www.earthsystemgrid.org/dataset/ucar.cgd.cesm2le.output.html
GPCC precipitation: https://www.dwd.de/EN/ourservices/gpcc/gpcc.html
BEST temperature: http://berkeleyearth.org/data/
and ERA5 SLP: https://www.ecmwf.int/en/forecasts/dataset/ecmwf-reanalysis-v5
Code used to create the Observational Large Ensemble and Dynamical Adjustment results are
publicly available at:
https://github.com/karenamckinnon/observational_large_ensemble/ and
https://github.com/terrayl/Dynamico, respectively.

**Author contributions**
CD led the overall effort and wrote the manuscript. ASP performed some of the calculations and
prepared the figures.



**Competing interests**

The contact author has declared that none of the authors has any competing interests.

**Acknowledgements**

We acknowledge the efforts of all those who contributed to producing the model simulations and observational data sets used in this study. We thank L. Terray for providing the dynamical adjustment results and K. McKinnon for providing the observational large ensemble results. The National Center for Atmospheric Research is sponsored by the National Science Foundation.

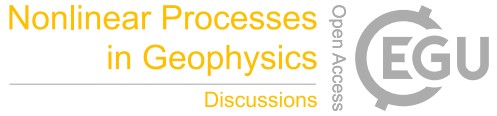

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
