# Peer review of "The Role of Internal Variability in Regional Climate Change"

_Nonlinear Processes in Geophysics, 2022_

## Author Response (AR1)

December 27, 2022

Dear Editor,

Please find below our point-by-point responses to the Reviewer and Contributed Comments on our manuscript "The Role of Internal Variability in Regional Climate Change" (NPG-2022-15) by Clara Deser and Adam S. Phillips, submitted as a Research article to the NPG Special Issue "Interdisciplinary perspectives on climate sciences – highlighting past and current scientific achievements". We very much appreciate the thoughtful, thorough and constructive comments from all of the Reviewers (official and otherwise), which has led us to improve the clarity of our presentation.

The Reviewers' original comments are included in plain black text, and our responses are given in blue italicized text beneath. Line numbers refer to those in the original manuscript.

Sincerely,
Clara Deser (corresponding author)
* * *
**RC1: Tamas Bódai**

This paper is very nicely written, with abundant information on its subject.

I would like to raise a caution about taking it for granted that any ensemble-statistics (E-stat') (even with a hypothetical infinite initial condition ensemble of a model) represents a forced response of the model climate. The following lines of the paper seem to have no such concern:

Because the temporal sequences of internal variability unfold differently in the
96 various ensemble members once the memory of the initial conditions is lost, one can estimate the
97 forced component at each time step (at each location) by averaging the members together, provided
98 the ensemble size is sufficiently large. The internal component in each ensemble member is then
99 obtained as a residual from the ensemble-mean.

However, Gabor Drótos and I (see reference below) worked out the conditions when we can regard an E-stat' (change) a sound quantifier of climate (change). This is a conditional definition of climate and it requires — for one thing — a time scale separation bw. certain fast and slow processes. Such a sound conditional climate change, however, might not be entirely forced, but the evolution of the slow system could introduce an unforced component. Thus, the concepts of climate change and forced change decouple. The following lines from the paper could be interpreted in our sense (not considering the citations), but then they would contradict the above quotation (l65-99):

In some 58 areas, climate trends driven by internal processes may even outweigh those due to anthropogenic
59 influences over the past 30-60 years (Deser et al., 2012, 2016 and 2017; Wallace et al., 2013; Swart
60 et al. 2015; Lehner et al. 2017).

*Thank you for alerting us to your new paper with Gabor Drótos, which we were not aware of. We fully agree that the presence of slow internal variations may confound identification of the forced response as the ensemble-mean change (Drobos and Bodai, 2022). We tried to account for this in our cited text (lines 96-99) with the clause "once the memory of the initial conditions is lost". Slow internal variations can still exist, but they will have different phases and amplitudes in the different ensemble members, so it seems to us that the ensemble-mean will still provide a reasonable estimate of the forced response after the memory of the initial-condition is lost. We also wish to point out that Section 6 of McKinnon and Deser (2018; also see related work in McKinnon and Deser, 2021) showed explicitly that high-frequency atmospheric variability as opposed to decadal timescale processes associated with slow ocean or coupled ocean-atmosphere modes dominates the internal component of 50-year trends in temperature and precipitation over Eurasia. A similar finding was reported in Deser et al. (2012), in which we explicitly compared the standard deviation of 56-year trends between the CCSM3 coupled model and the CAM3 atmosphere-land model (without ocean variability) and found that the two distributions were not significantly different over most of the NH continents (see their Fig. 9), attesting to the dominant influence of (red or white noise) atmospheric circulation variability as opposed to slow decadal processes associated with the ocean or coupled ocean-atmosphere system. To address your comment in our revised manuscript, we have added the following sentence after lines 58-60:*

*"It is important to note that such internally-generated multi-decadal trends need not originate from slow processes within the ocean or coupled ocean-atmosphere system: indeed, random fluctuations of the atmospheric circulation independent of oceanic influences have been shown to drive a large fraction of long-term precipitation and temperature trends over North America and Eurasia (Deser et al. 2012; McKinnon and Deser, 2018)."*

On the subject of "...a larger ensemble may be needed for some aspects of the forced response than others" (l99-100), maybe you wish to cite my papers that advocate that cross-correlations of TWO quantities as opposed to some E-stat' of ONE observable quantity tend to show much smaller SNR. So, forced changes of teleconnections can be difficult to detect even in an ensemble. Also, typically the higher the statistical momentum/quantile, the smaller the SNR.

*Thank you. In our revised version, we now cite your papers regarding the need for larger ensemble sizes for quantities with lower SNR such as higher statistical moments/quantiles and teleconnections (cross correlations between two quantities).*

I didn't quite understand how the obs ensemble can help in re-assessing the detectability of trends in single observed realisations (paragraph starting with line 376). The model ensemble can have a bias in the forced trend because of model error (or changes of the slow system, as mentioned above, or numerical model drift, being an artefact). Isn't the forced trend of the OBS

ensemble the same as that of the model ensemble by construction, i.e., possibly biased? I don't see a solution for this problem.

*Yes, in the case of Fig. 8 to which this paragraph pertains, the forced trends are the same as that of the model ensemble by construction. Our point here is to quantify model biases in the noise (internal variability of trends as measured by the standard deviation of trends across ensemble members) and how they affect the SNR. In our revised manuscript, we have added the text in bold italics to the sentence in lines 377-378: "We address this question by using the OBS LE $\sigma$ values in place of the model's $\sigma$ values in the signal-to-noise calculation **(note that the "signal" in the two LEs is identical by construction)**".*

l553 this is the first time?

*Yes, this is the first time to the best of our knowledge. We have rewritten the text to emphasize this point.*

l581 "wet" and "dry" — check for the consistency of the directionality of double quotation marks

*Thank you, corrected.*

l605 combined the internal variability?

*Thank you, changed as suggested.*

Note: I do not make recommendation to editors for or against publishing a paper. I selected "minor revision" only to be able to submit my review; please consider it void.

Tamas Bodai

References

Gabor Drótos, Tamas Bódai 2022. On defining climate by means of an ensemble [Preprint]. https://essoar.org (2022) https://doi.org/10.1002/essoar.10510833.2

Tamás Bódai, June-Yi Lee, Aneesh Sundaresan.(2022) Sources of Nonergodicity for Teleconnections as Cross-Correlations, Geophysical Research Letters, 49, 8, e2021GL096587, doi: 10.1029/2021GL096587

Bódai, T., G. Drótos, M. Herein, F. Lunkeit, and V. Lucarini (2020) The Forced Response of the El Niño–Southern Oscillation–Indian Monsoon Teleconnection in Ensembles of Earth System Models. J. Climate, 33, 2163–2182, https://doi.org/10.1175/JCLI-D-19-0341.1
* * *
**CC1: Gabor Drobos**

The preprint undoubtedly contains interesting and important information regarding the detectability and properties of regional climate change. The result that thermodynamic-residual trends match closely the corresponding ensemble-mean trends is particularly remarkable. Even though most of the conclusions are well established or are presumably sufficiently robust, there is a number of issues, as I see, that require correction or further thought before final publication.

*Thank you for your thorough set of comments and overall positive assessment of our study.*

The most important one is a factual error in the text, which has a direct implication for the interpretation of the results on the signal-to-noise ratio. In lines 362-365, it is stated that "For a normal distribution, a signal-to-noise ratio greater than two indicates that the ensemble-mean (forced) trend is significantly different from zero at the 95% confidence level: that is, there is less than a 5% chance that the ensemble-mean trend could have been a result of random internal variability." It is easy to demonstrate that this statement is wrong in the sense that the actual confidence level is higher under the assumption of a Gaussian distribution. The standard deviation of the trends, which appears in the denominator of the signal-to-noise ratio, is (at least approximately and apart from sampling uncertainty) uniquely determined by the so-called natural probability measure (see Drótos et al., 2015; Tél et al., 2020) and has a finite value, irrespective of the number of ensemble members used. On the other hand, by increasing the number of ensemble members, the ensemble mean of the trends can be determined with arbitrary precision: that is, the chance that a nonzero ensemble mean is obtained while the true expectation value of the trends is zero can be arbitrarily reduced. In the particular case when the ensemble mean happens to be twice the standard deviation, the chance that the true expectation value of the trends is zero can thus be less than 5%: actually, it can be arbitrarily small if the number of ensemble members is sufficiently large (irrespective of whether the trends are distributed according to a Gaussian).

In fact, if the number N of ensemble members is sufficiently large, then the sampling distribution of the ensemble mean of the trends (as an estimator of the true expectation value of the trends) is a Gaussian that is centered on the true expectation value and has a standard deviation that scales as 1/sqrt(N), according to the central limit theorem. Under the null hypothesis that the true expectation value is zero, the task is to find the value above which (in an absolute sense) this Gaussian integrates to the desired significance level (one minus confidence level). For instance, the value sought is twice the standard deviation for a 5% significance level. If we furthermore assume that the parent distribution (that of the trends observable in the individual ensemble members) is a Gaussian with a standard deviation estimated precisely by the sample standard deviation σ of the trends computed over the ensemble, the sampling distribution of the ensemble mean will have a standard deviation of σ/sqrt(N). In this case, the signal-to-noise ratio, which is associated with a given significance or confidence level and is defined by dividing the actual ensemble mean by the standard deviation of the sampling distribution, will be sqrt(N) times higher than what is presented in the preprint. For the 100-member CESM2 LE, it means a 10-fold (!) increase with respect to the presented results. It should be emphasized, however, that the assumptions about the shape and standard deviation of the parent distribution may not at all be

justified, so that the results shown in the preprint may only provide a qualitative guidance in the absence of a dedicated investigation.

*Thank you for pointing out the confusion between the standard error of the mean and the standard deviation. We realize that we did not convey our intent properly. We are not interested in whether the ensemble-mean trend is statistically significant relative to the spread of trends across the individual members of the ensemble, in which we case we would use the standard error of the mean. Rather, we want to know how large (in an absolute sense) the ensemble-mean trend is compared to the internal component of trends as sampled by the CESM2 LE. In other words, how likely is it that the ensemble-mean trend could be overwhelmed by the internal trend in any given realization of the LE? For this purpose, a useful metric is the relative amplitude of the ensemble-mean trend compared to a typical (one standard deviation) internal trend: e.g., an absolute SNR>1 would signify that the forced trend is greater than a typical internal trend, and an absolute SNR>2 would signify that the forced trend is twice as large as a typical internal trend, and regions where the absolute SNR<1 would mean that a typical internal trend amplitude is greater than the forced trend. We have rewritten the text on lines 362-365 to make this point, and removed the confusing language about whether the ensemble-mean trend is statistically significant. Here is our rewritten text:*

*"This "signal-to-noise" ratio provides a metric of the likelihood that the ensemble-mean (e.g., forced) trend might be overwhelmed by the internally-generated trend in any given ensemble member (and by extension, the real world). Assuming that the 100-member set of 50-year trends follows a normal distribution (not shown, but see related results in Deser et al. 2012; Thompson et al. 2015; Deser et al. 2020), a signal-to-noise ratio greater than one (two) indicates that the magnitude of the ensemble-mean (forced) trend is larger than (more than twice as large as) that of a typical (e.g., one standard deviation) internal trend, and a signal-to-noise ratio less than one indicates that the amplitude of a typical internal trend exceeds the magnitude of the forced trend."*

I would continue my comment with an overarching issue regarding the terminology. Already the title suggests that internal variability can have an effect on climate change, and this is explicitly confirmed by the first sentence of the short summary, as well as referring to accuracy in its last sentence. As Tamás Bódai has already pointed out in RC1 of the discussion about the preprint, this notion is only meaningful if climate is defined to be conditional on the state of slower system components which have their "own" internal variability and can thus possibly induce unforced changes in the probability distribution that defines climate, in the sense discussed in Drótos and Bódai (2022). But 50-year trends, analyzed in the preprint, or those of similar length, can hardly be dominated by such changes; instead, they mostly originate from processes having decadal time scales and sufficiently rapid forced changes. Even if the effect of variations in slower system components is not negligible, it may (and hopefully does) remain unique for some time; in any case, the differences in the mentioned trends between the individual members of the ensemble are mostly due to faster processes of the Earth system. Therefore, these differences should definitely not be interpreted as differences in the pace of climate change, at least if the particular study targets the time scale of a century (still see Drótos and Bódai, 2022). On the contrary, if slower system components do not deteriorate uniqueness on the time scale in question, these differences should be regarded as an inherent property of a single (but changing)

climate. As a consequence, writing about internally driven or non-unique "climate trends" in this context (lines 58 and 137), as if the pace of climate change were (substantially) dependent on the particular realization, would be safer to avoid (I use the word 'substantially' to refer to a potential non-unique effect of slower system components).

*Thank you for this comment. We did not intend to suggest that the pace of forced climate change is (substantially) dependent on the particular realization. We simply meant to convey that the superposition of internal variability, which is unique to each realization after initial-condition memory is lost, and forced climate change, which is common to each realization, results in a different trend magnitude in each realization. In our revised manuscript, we have rewritten the text in lines 137-139 to clarify our meaning and removed the problematic terminology "are not unique".*

*In response to your comment, we have also modified the title to be more accurate and informative as follows: "A Range of Outcomes: The Combined Effects of Internal Variability and Anthropogenic Forcing on Regional Climate Trends".*

*Regarding your statement: "But 50-year trends, analyzed in the preprint, or those of similar length, can hardly be dominated by such changes; instead, they mostly originate from processes having decadal time scales and sufficiently rapid forced changes", we respectfully disagree with the text after the semi-colon. As we showed explicitly in Section 6 of McKinnon and Deser (2018; also see related work in McKinnon and Deser, 2021), high-frequency atmospheric variability as opposed to decadal-timescale processes associated with slow ocean or coupled ocean-atmosphere modes dominates the internal component of 50-year trends in temperature and precipitation over Eurasia. A similar finding was reported in Deser et al. (2012), in which we explicitly compared the standard deviation of 56-year trends between the CCSM3 coupled model and the CAM3 atmosphere-land model (without ocean variability) and found that the two distributions were not significantly different over most of the NH continents (see their Fig. 9), attesting to the dominant influence of (red or white noise) atmospheric circulation variability as opposed to slow decadal processes associated with the ocean or coupled ocean-atmosphere system. To convey this information in the revised manuscript, we have added the following text to the end of the $3^{rd}$ paragraph in the Introduction:*

*"It is important to note that such internally-generated multi-decadal trends need not originate from slow processes within the ocean or coupled ocean-atmosphere system: indeed, random fluctuations of the atmospheric circulation independent of oceanic influences have been shown to drive a large fraction of long-term precipitation and temperature trends over North America and Eurasia (Deser et al. 2012; McKinnon and Deser, 2018)."*

*McKinnon, K. A and C. Deser, 2018: Internal variability and regional climate trends in an Observational Large Ensemble. J. Climate, **31**, 6783–6802, doi:10.1175/JCLI-D-17-0901.1.*

*McKinnon, K. A. and C. Deser, 2021: The inherent uncertainty of precipitation variability, trends, and extremes due to internal variability, with implications for Western US water resources. J. Climate, **34**, 9605-9622, doi: 10.1175/JCLI-D-21-0251.1.*

*Deser, C., A. S. Phillips, V. Bourdette, and H. Teng, 2012: Uncertainty in climate change projections: The role of internal variability. Climate Dyn., **38**, 527-546, DOI 10.1007/s00382-010-0977-x.*

The sentence in line 166 seems to be problematic from the same point of view; and even though they are widely used, the expressions "anomalous climate event" (line 62), "climate anomaly" (lines 182, 192 and 594) and "climate extreme" (line 590) appear to suffer from a similar conceptual issue. (These latter expressions sound as if climate could be anomalous or extreme at a given time within a single realization mostly due to internal variability --- this would only be meaningful if internal variability in slower processes, with time scales beyond the targeted one, induced these anomalies and extremes.) Also, I wouldn't advise writing that internal variability (substantially) "limits the accuracy of climate model projections" on time scales longer than a decade but not longer than a century [line 18; climate projections are usually meant to be "uninitialized" (section 11.1 of Kirtman et al., 2013) and thus fully encompass the statistics of the internal variability of the faster processes at least] or generates (substantial) "uncertainty" in them (lines 50 and 53); instead, internal variability (of the faster processes at least) represents an inherent property of climate and thus its projections. A related remark is that the ensemble mean of the trends obtained in individual members is principally interesting for the purpose of comparison with instrumental observations (having an eye on detection and attribution). If the aim were to quantify the effects of forcing or slower system components on climate, I believe that it would be more useful to investigate the trends of the ensemble mean (or those of further statistical quantifiers evaluated with respect to the ensemble).

*We agree that internal variability represents an inherent property of climate and thus its projections. However, we are trying to make the point that projections in the IPCC are typically represented by only the forced climate change component, which ignores the fact that internal variability will also play a role in the actual outcome. Thus, we prefer to keep the text about "internal variability limits the accuracy of climate model projections", since projections in our view should include both the forced component and the unforced (and largely unpredictable) component.*

*In the revised manuscript, we have made the following changes in response to your comments:*
- *Lines 17 and 19: changed "climate changes" to "climate trends" for clarity*
- *Line 18: removed the phrase "on timescales longer than a decade" and replaced it with "on long timescales".*
- *Line 62: removed the expression "anomalous climate event" and replaced it with "extreme weather events".*
- *Line 182: removed "surface climate anomalies" and replaced it with "observed temperature and precipitation data".*
- *Line 192: removed "surface climate anomalies" and replaced it with "temperature or precipitation".*
- *Line 594: removed "climate anomalies" and replaced it with "temperature and precipitation data".*
- *Line 590: removed "climate extreme".*

*- Title: Changed to reflect your comments about clarity as follows: "A Range of Outcomes: The Combined Effects of Internal Variability and Anthropogenic Forcing on Regional Climate Trends over Europe".*

Having mentioned the possibility of unforced changes induced by slower system components, I would point out that climate can be easily defined only if these unforced changes remain unique during the time span of the study. As mentioned above, hopefully this is the case, but whether or not this is actually so, such unforced changes may appear in ensemble statistics with some weight, which is a problem already discussed by Tamás Bódai in RC1 and RC2. In such a case, variations in ensemble statistics do not entirely represent a forced response, as opposed to what is stated in line 97 and made use of throughout the text.

*Please see our response to Tamás Bódai's comment above (RC1).*

I list further substantial issues in the order as they appear in the preprint.
lines 97-98: The question of separating forced change and internal variability seems to be simplified here to determining the time evolution of the ensemble mean and taking the differences from the ensemble mean in individual ensemble members. However, internal variability is characterized by a full probability distribution the time evolution of which (as a forced response, or perhaps including an unforced component originating from slow processes, too) concerns all statistical quantifiers, as actually acknowledged in lines 102 and 108.

*No changes needed.*

175-177: They are not only decadal shifts in regional anthropogenic aerosol emissions that violate the assumption of a slow forced change, but also greenhouse gas concentrations and solar activity can substantially vary on decadal time scales, and volcanic eruptions have an instantaneous and sometimes very strong impact.

*Agreed; text has been revised accordingly.*

207: It depends on the choice of variable if the memory of the initial state can become negligible by the time specified. There are system components, e.g., the deep ocean, for which the statement is not true.

*Agreed; text has been revised accordingly.*

215 and 554-556: The realizations of internal variability in the OBS LE were obtained in McKinnon and Deser (2018) under the assumption that the forced response is described by the CESM1 LE. In particular, the $\beta$ coefficients and the $\varepsilon$ residuals of Eq. (1) of McKinnon and Deser (2018) were obtained using ordinary least squares regression under this assumption. Therefore, it appears to me that the trends in the realizations of internal variability in the OBS LE are ensured to be consistent with observations only if the full trends are obtained by adding the forced trend of the CESM1 LE to the internal trend of each OBS LE member; using the forced trend of the CESM2 LE [or the CMIP5 ensemble, as in McKinnon and Deser (2018)] or the

thermodynamic-residual trend (obtained from dynamical adjustment) for the same purpose might yield spurious results. This issue may affect the corresponding analyses throughout the preprint.

*The β coefficients and the ε residuals of Eq. (1) of McKinnon and Deser (2018) for the OBS LE were obtained after removing the component of variability linearly related to the CESM1 LE ensemble-mean global-mean temperature timeseries following the method of Dai et al. (2015). Then, the ensemble-mean trends from the CESM1 LE were added back to the internal trends of the OBS LE to produce the "full" OBS LE trends. Here, we have subtracted the CESM1 LE ensemble-mean trends from the "full" OBS LE trends and then added the CESM2 LE ensemble mean trends to produce the results shown in the paper. Since the shape of the ensemble-mean global-mean temperature timeseries in the CESM2 LE is very similar to that of the CESM1 LE over the historical period (not shown), we do not expect the use of the former rather than the latter in our OBS LE methodology to substantially affect our results. Regarding your point about whether this introduces spurious results when we use the thermodynamic-residual trend obtained from dynamical adjustment as our estimate of the forced trend, we agree that it would be preferable to re-compute the OBS LE β coefficients and ε residuals after removing the variability linearly related to the observed global-mean temperature timeseries based on dynamical adjustment for consistency. However, this is well beyond the scope of our study and is left to future work. We feel that mentioning these somewhat subtle points in the manuscript would require considerable explanation and digression, and detract from our intentional "big picture" framing of the OBS LE. We are embarking on a new study in which we will investigate these points in detail – please stay tuned.*

*Dai, A., J. C. Fyfe, S.-P. Xie, and X. Dai, 2015: Decadal modulation of global surface temperature by internal climate variability. Nat. Climate Change, 5, 555–559, https://doi.org/10.1038/ nclimate2605.*

335-350 and Fig. 7: The respective results for σ could be interesting for the future (2022-2071) trends as well in the CESM2 LE, even if a comparison with observations is not possible.

*Yes, we thought about including the σ maps for the future (2022-2071) trends in Fig. 7 but decided it was not essential to this study.*

355: "significantly" should be replaced by "significant". More importantly, the particular statistical test should be specified. Actually, it should also be demonstrated that the conditions for the applicability of the given test are met.

*Thank you for pointing out our typo; it has been corrected. The test is an f-test, and is now be mentioned in the revised caption.*

463: While the 5th-to-95th percentile range indeed narrows slightly, the 25th-to-75th percentile range appears to narrow even more in a relative sense, which would be worth mentioning in the text, I believe.

*Good point. This is now mentioned in the revised text.*

604: What is referred to as "the model's forced thermodynamic trend" is in fact not purely thermodynamic (by construction, it includes changes in circulation however minuscule they are), and its purely forced nature is also questionable (as discussed earlier in relation to the effect of slower system components). It would be a more cautious choice to simply write "the model's ensemble mean trend".

*We prefer to keep the existing text, since we have confirmed that the contribution of forced dynamics is negligible (see our response to RC4 below) and that the contribution of slower system (internal) components to the ensemble-mean trend is also negligible as discussed above in our response to an earlier comment.*

Finally, I would mention two technicalities that could facilitate comprehension and reproducibility:
- It could be explicitly stated that panels (a) and (c) of Figs. 10 and 11 are identical to OBS and EM in Figs. 1 and 2, respectively, except that contours of SLP trends are also included.

*Thank you; done.*

- The serial number of the ensemble members used in Figs. 10b, 10d, 10f, 10h, 11b, 11d, 11f and 11h could be specified.

*We thought about this, but decided to omit the ensemble members as it clutters the figure.*

In spite of the several critical comments, I do believe that the results presented in the preprint are important and will be useful for future research.

*Thanks again for your thorough and constructive assessment.*

References:
G. Drótos, T. Bódai and T. Tél (2015). "Probabilistic concepts in a changing climate: A snapshot attractor picture". J. Climate 28, 3275–3288. https://doi.org/10.1175/JCLI-D-14-00459.1

G. Drótos and T. Bódai (2022). "On defining climate by means of an ensemble". ESSOAr (preprint). https://doi.org/10.1002/essoar.10510833.3 [Note the update on 2022-11-07.]

B. Kirtman, S.B. Power, J.A. Adedoyin, G.J. Boer, R. Bojariu, I. Camilloni, F.J. Doblas-Reyes, A.M. Fiore, M. Kimoto, G.A. Meehl, M. Prather, A. Sarr, C. Schär, R. Sutton, G.J. van Oldenborgh, G. Vecchi and H.J. Wang (2013). "Near-term Climate Change: Projections and Predictability". In: "Climate Change 2013: The Physical Science Basis. Contribution of Working Group I to the Fifth Assessment Report of the Intergovernmental Panel on Climate Change" [Stocker, T.F., D. Qin, G.-K. Plattner, M. Tignor, S.K. Allen, J. Boschung, A. Nauels, Y. Xia, V. Bex and P.M. Midgley (eds.)]. Cambridge University Press, Cambridge, United Kingdom and New York, NY, USA.
https://www.ipcc.ch/site/assets/uploads/2018/02/WG1AR5_Chapter11_FINAL.pdf

T. Tél, T. Bódai, G. Drótos, T. Haszpra, M. Herein, B. Kaszás and M. Vincze (2020). "The theory of parallel climate realizations: A new framework of ensemble methods in a changing climate - an overview". Journal of Statistical Physics 179, 1496–1530. https://doi.org/10.1007/s10955-019-02445-7

K. A. McKinnon and C. Deser (2018). "Internal Variability and Regional Climate Trends in an Observational Large Ensemble". Journal of Climate 31, 6783-6802. https://doi.org/10.1175/JCLI-D-17-0901.1
* * *
**RC4 Referee #2**
REVIEW FOR "The Role of Internal Variability in Regional Climate Change", by Clara Deser and Adam S. Phillips, submitted to Nonlinear Processes in Geophysics

Summary:

The authors analyse historical and future European temperature and precipitation trends in the CESM2 Large Ensemble, observations, and an "Observational Large Ensemble". The analysis is based on an analogue-based dynamical adjustment method, which is used to disentangle dynamical (based on SLP analogues) and thermodynamical (residual) trends in the climate model and observational ensemble. As a result, the authors show that internal climate variability is a crucial source of uncertainty in future European climate, the level of which is broadly comparable in magnitude between the Obs-LE and CESM2-LE, and that the thermodynamical component of observations agrees well with the forced CESM2-LE component for temperature, and less well for precipitation.
Overall, the paper provides a very useful illustration of internal variability in present and future European climate, and a constructive discussion of current and outstanding issues in dynamical adjustment. The paper is also very well written and logically structured. I still have a few concerns that are outlined below, however, and I would therefore recommend moderate revisions.

*Thank you for your favorable assessment and constructive comments and suggestions.*

Major issues:
(1) Abstract

The Abstract is well-written, however it is somewhat disconnected from the actual analysis conducted in the paper. At present the Abstract reads a bit like that from a Perspective paper, while the (by far) largest part of the paper presents actually a specific analysis of European climate. Hence, I would recommend to adjust the Abstract such that it (also) reflects the analysis conducted in the paper.

*Agreed!  We have adjusted the Abstract to better reflect the content of the paper as suggested. The new Abstract is now nearly doubled in length at 198 words, but still within the word count limit of 200 words.*

(2) Implications of high climate sensitivity in CESM2 for interpretation of thermodynamical trends

The authors interpret "the good agreement between the observed thermodynamic-residual trend component and the model's forced thermodynamic trend" (l. 604) as "further underscoring the realism of CESM2" (l. 605), and that "the model's forced temperature trend is realistic" as a powerful conclusion (l. 521). This conclusion is based on the temperature dynamical adjustment discussed on p. 30, where the authors argue that "observed thermodynamic trend is much closer

in amplitude (and arguably pattern) to the model's forced response".

While I agree that these results are in general really encouraging, I do think that some caution is warranted: CESM2 is known for high climate sensitivity, so (I believe) we *should* expect some discrepancy in the amplitude of the pattern, and -contrariwise- a higher similarity in the pattern itself. Hence, why is the observed thermodynamical pattern's amplitude over Europe so high as to even match that of a high climate sensitivity model?

*Although CESM2 is a "high climate sensitivity" model, its global mean temperature rise over the historical period (at least since 1920 or 1950) is quite similar to observations as shown in the plot below. In this plot, the gray curve is HadCRUTv5 and the blue curve is the CESM2 LE ensemble mean; the light blue (dark blue) shading is the 25th-75th (5th-95th) percentile range across the LE ensemble members.*

[Figure]

*Fig. R1. Global mean temperature anomaly timeseries relative to the 1901-2012 climatology for observations (HadCRUTv5; grey curve) and the CESM2-LE ensemble mean (blue curve). The light blue (dark blue) shading is the 25th-75th (5th-95th) percentile range across the CESM2-LE ensemble members. The number shown in the lower left of the plot (80%) refers to the fraction of the time that the observed curve lies outside the 5th-95th percentile range of the CESM2-LE ensemble. Graphic produced by the Climate Variability Diagnostics Package for Large Ensembles (https://www.cesm.ucar.edu/projects/cvdp-le).*

*This graph shows that the "high climate sensitivity" of CESM2 LE is not a feature of the historical period that we analyse (1972-2021). We note that the moniker "high climate sensitivity" refers specifically to the Equilibrium Climate Sensitivity (ECS) of the model as diagnosed by means of CESM2 simulations in a slab-ocean configuration in response to an instantaneous doubling of CO2 (Gettelman et al. 2019). Evidently, a high ECS does not necessarily translate to a high transient climate sensitivity over the historical record. In our revised manuscript, we now mention this important point by adding the following text after line 523:*

*"It may seem surprising that the model's forced temperature trend agrees so well in amplitude with the observed thermodynamical-residual trend, given that CESM2 has been characterized as a "high climate sensitivity" model (Gettelman et al., 2019). However, this characterization*

*refers specifically to the model's equilibrium climate sensitivity (diagnosed as the model's response to an instantaneous doubling of CO2 based on a slab-ocean configuration), and does not translate to a high transient climate sensitivity over the 1962-2021 period of record analyzed here, as evidenced by the fact that the observed global-mean temperature increase lies within the ensemble-spread of global-mean temperature trends simulated by the CESM2-LE for this time period (not shown)."*

*Gettelman, A., Hannay, C., Bacmeister, J. T., Neale, R. B., Pendergrass, A. G., Danabasoglu, G., et al. (2019). High climate sensitivity in the Community Earth System Model Version 2 (CESM2). Geophys. Res. Lett., 46, 8329– 8337. https://doi.org/10.1029/2019GL083978*

Moreover, for precipitation more careful conclusions would be warranted, as the residual component does not closely resemble the model's forced response. For example, the authors attribute the (large) pattern disagreement in Central Europe to "lower signal-to-noise" found in this region compared to other areas (l. 544-546), and further pattern disagreement over large areas in South Europe, such as the Balkans, Turkey, and Italy is only briefly mentioned. Here, I believe it would benefit the discussions if the authors would discuss this a bit more in-depth, and explain where the "lower signal-to-noise" explanation in this region comes from (because this is a transitional region between southern drying and northern wettening?).

*We agree, and have rewritten this sentence to more comprehensively describe the areas of disagreement in sign and the reasons for it as follows:*

*"Notable areas of agreement in the sign of trends include drying over most of Spain, Portugal, Algeria, Turkey and Syria, and wetting over parts of northern and north-central Europe; disagreement in sign is found over many central European countries (France, Germany, Switzerland, Austria, Ukraine, Romania and southern Russia) where the signal-to-noise is low (Figs. 8d,e) due to a combination of low signal in the transition region between southern drying and northern wetting (Fig. 11c) and high noise (Figs. 7d,e). The low signal and high noise in these areas limits the accuracy of the dynamical adjustment results, where the error of the method is of the same amplitude as the thermodynamic-residual trend (see Guo et al., 2019 for details)."*

(3) The authors assume that the forced CESM2 trend (i.e., ensemble average) reflects the thermodynamical response to climate change. This is consistent with literature, but reflects some simplification, which the authors acknowledge in their discussion. But, the authors also say that "future trends in SLP also contain a modest forced component indicative of enhanced westerlies over the continent" (l. 479), and I believe there may be in addition nuanced forced dynamical components with only a modest SLP signature.

In earlier literature (Deser et al. 2016), the authors actually use their dynamical adjustment method to show in their Fig. 7 that the average across the dynamical contribution is rather small. I believe it may possibly benefit the present paper and argument to include and discuss a similar figure for Europe?

*As suggested, we have performed a new dynamical adjustment calculation where we apply the method to the observed SLP anomaly field after removing the forced SLP component determined from the ensemble-mean of the CESM2-LE SLP anomaly field at each time step (where the term "anomaly" indicates that the monthly climatology was first removed from observations and from each member of the CESM2-LE). The resulting dynamical and thermodynamic-residual contributions to the observed temperature and precipitation trends show only minor differences with those based on the full observed SLP anomaly field, as expected given the lack of appreciable amplitude of the forced SLP trend. For the Reviewer's benefit, we show these results in the figure below, but do not think it is worthwhile to include them in the manuscript.*

[Figure]

*Fig. R2. As in Figs. 12 and 13 in the manuscript, except that the model's forced SLP has been removed from the observed SLP in the dynamical adjustment calculation as described above.*

Minor issues:

l. 150. McKinnon et al. 2017 is missing in the references section

*Thank you; missing reference has been added.*

l. 631. The connection to predictability studies and the "signal-to-noise paradox" is interesting, but the short discussion is hard to follow. Maybe the implications could be made a bit more explicit here.

*Thank you; we have rewritten the discussion to make it easier to follow.*

References:

Deser, C., Terray, L. and Phillips, A.S., 2016. Forced and internal components of winter air temperature trends over North America during the past 50 years: Mechanisms and implications. Journal of Climate, 29(6), pp.2237-2258.

---

## Author Response (AR2)

January 30, 2023

**Re: npg-2022-15**
Title: A Range of Outcomes: The Combined Effects of Internal Variability and Anthropogenic Forcing on Regional Climate Trends over Europe
Author(s): Clara Deser and Adam S. Phillips
MS type: Research article
Iteration: Correction
Special Issue: Interdisciplinary perspectives on climate sciences – highlighting past and current scientific achievements

Dear Editor,

We have corrected the typos in the text as requested by the Editor.

Since our manuscript was selected as highlight paper, here is our 486 character short summary:

"Disentangling the effects of internal variability and anthropogenic forcing on regional climate trends remains a key challenge. Due to its largely unpredictable nature on timescales longer than a decade, internal climate variability limits the accuracy of climate model projections and complicates efforts to attribute past climate trends and evaluate climate models. In this study, we review and synthesize recent advances in climate modeling and atmospheric dynamics that have led to novel insights on these issues, with illustrative examples for European climate."

Sincerely,
Clara Deser